

# On the intrinsic pinning and shape of charge-density waves in 1D Peierls systems

**Olivier Cépas and Pascal Quémerais**

Université Grenoble Alpes, CNRS, Grenoble INP, Institut Néel, 38000 Grenoble, France

## Abstract

Within the standard perturbative approach of Peierls, a charge-density wave is usually assumed to have a cosine shape of weak amplitude. In nonlinear physics, we know that waves can be deformed. What are the effects of the nonlinearities of the electron-lattice models in the physical properties of Peierls systems? We study in details a nonlinear discrete model, introduced by Brazovskii, Dzyaloshinskii and Krichever. First, we recall its exact analytical solution at integrable points. It is a cnoidal wave, with a continuous envelope, which may slide over the lattice potential at no energy cost, following Fröhlich's argument. Second, we show numerically that integrability-breaking terms modify some important physical properties. The envelope function may become discontinuous: electrons form stronger chemical bonds which are local dimers or oligomers. We show that an Aubry transition from the sliding phase to an insulating pinned phase occurs when the model is no longer integrable.

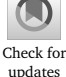

# 1 Introduction

In 1955, the instability of a simple linear metallic chain of equidistant atoms was predicted by Peierls [1, 2]. Below a critical temperature, called Peierls temperature, a new state emerges which is characterized by a lattice distortion associated to an electronic charge-density modulation. This state is called a charge-density wave (CDW) state. The current CDW theory is essentially based on the Fröhlich model [3] where the Hamiltonian of an electron gas is perturbed by an electron-lattice interaction which is *linear* with the lattice distortion (for particularly comprehensive reviews, see Refs. [4, 5]). The Peierls transition then results from the energy lowering of the occupied electronic states below the Fermi level which exceeds the energy cost of the elastic distortion at low temperatures. Linear response approximation for the electrons has been extensively used in this theory: the static electronic Lindhard susceptibility of a one-dimensional metal $\chi(q, \omega = 0)$, calculated at first-order in perturbation theory, shows a peak at $q_0 = 2k_F$, $k_F$ being the Fermi wavevector [6]. An atomic position modulation $\delta x_{q_0}$ with wavevector $q_0$ then induces a proportional modulation of the electronic charge density

$\delta\rho_{q_0} = g\chi(q_0, 0)\delta x_{q_0}$, where $g$ is the so-called electron-phonon coupling constant. The electronic energies close to the Fermi level are modified and an electronic gap opens up due to the perturbation induced by the lattice modulation. The total energy is minimized with respect to the distortion parameter $\delta x_{q_0}$, also fixing the charge modulation $\delta\rho_{q_0}$ and the electronic gap. Below the Peierls temperature, $\delta x_{q_0}$ and $\delta\rho_{q_0}$ are non-vanishing and the CDW state is stabilized [4,5]. The resulting modulation of the position of atom $i$ along the chain and its electronic charge are given by

$$\delta x_i \sim \delta x_{q_0}\cos(q_0 i a + \phi), \tag{1}$$

$$\delta\rho_i \sim \delta\rho_{q_0}\cos(q_0 i a + \phi), \tag{2}$$

where $a$ is the interatomic distance and $\phi$ a phase. One sees that when the parameter $\frac{q_0 a}{2\pi}$ is a rational number, the charge modulation $\delta\rho_i$ and the lattice distortion $\delta x_i$ repeat after a period. The chain has a periodicity and the CDW is said to be commensurate. Reversely, when $\frac{q_0 a}{2\pi}$ is an irrational number, the chain has no periodicity and the CDW is incommensurate. It was suggested by Fröhlich [3] that in the incommensurate case, the total energy does not depend on the phase $\phi$. In other words, there is a continuous degeneracy of the ground-state with respect to the phase, and the charge and lattice modulation may move freely along the chain: the CDW is *not pinned* by the lattice, it can *slide* freely. One says that the system has a Fröhlich conductivity. Reversely, in the commensurate case, this continuous degeneracy is lost and the CDW is *pinned* by the lattice [7].

Most discussions on CDW systems, in one or two dimensions, are based on the above linear response scheme. Further approximations may ameliorate the theory, for example by replacing the Lindhard susceptibility by the generalized susceptibility obtained within the random phase approximation in order to include Coulomb interactions [8]. Other authors have also developed strong electron-phonon coupling theories [8–11], especially for two-dimensional CDW systems. However, these developments remain basically in the same perturbative framework for the electrons and the results are not fundamentally modified: the CDW state results from the instability of a metal, as proposed by Peierls.

Furthermore, whether or not the Peierls-Fröhlich theory applies to experiments remains a controversial issue. Indeed, there is now a profusion of materials undergoing CDW transitions, where the theory can be tested. Beside well-known quasi-one-dimensional (1D) compounds [12,13] such as the trichalchogenides of transition metals ($NbSe_3$ for example), many 2D and even 3D CDWs exist. Notably, among the dichalchogenides, the commensurate/incommensurate 2D CDW states in $1T$-$TaS_2$, $2H$-$TaSe_2$ and $1T$-$TiSe_2$ were discussed by Rossnagel [14]. Based on detailed experimental ARPES data, he was able to show that it is the lowering of all occupied electronic bands which stabilizes the CDW, not only the lowering of the electronic state energies close to the Fermi level. Similarly, square nets of chalcogenides (Se or Te) are present in many binary, ternary and quaternary compounds [15, 16]. All these square nets undergo commensurate or incommensurate Peierls distortions which are described in terms of various polychalcogenide oligomers formation in the nets [15–19]. Here again, as noted by Patschke [16], the electronic band calculations show that the CDW stabilization is dominated by low-lying electronic levels. Another striking example has been recently provided by Gaspard [20]. Many covalent materials of columns 14-16 elements (examples are As, Se, Te, Br, I) develop 3D networks of successive short and large atomic distances. These structures appear as Peierls distortions from an unstable cubic structure. Yet, the electronic energy gain involved in these distortions goes well beyond the simple perturbation of the levels close to the Fermi level. A last challenging example is provided by the now common observations of 2D or 3D, static or fluctuating, CDW orders in Cu-based high-$T_c$ superconductors [21–25]. Although it is an ubiquitous phenomenon, the origin of these CDW orders remains elusive [26]. It however enlightens and renews the possibility of electron-based mechanisms (*i.e.* with an

electronic "glue") of superconductivity in cuprates [27–30].

Such observations contradict somewhat the simplistic Peierls-Fröhlich linearized framework. In reality, in order to construct a more general theory of CDW, one needs first to consider models with *nonlinear* electron-lattice interactions. Any realistic model is, by essence, nonlinear. The minimal number of ingredients to describe the basic chemical situation is then three. There is first an attractive term which gives a resonant bonding state for the electrons. It decreases with the interatomic distance in a nonlinear way. Secondly, one needs a repulsive part to avoid the collapse of the structure. These first two terms explain the chemical bonding of atoms in a standard way. Finally, a pressure term is added which may represent the effect of a real mechanical pressure, or may model a chemical pressure induced by the surrounding atoms. All these ingredients form *discrete* models since any atomic structure is discrete. These models have to be studied nonperturbatively, in order to find both the distortions and the electronic energies self-consistently and without expanding *a priori* around a putative metallic state.

In the present paper, we study such a nonlinear model in 1D, which was introduced by Brazovskii, Dzyaloshinskii and Krichever [31]. We call it the BDK model and its definition is given in section 2. In their original paper, the authors have shown that for special values of the model parameters, the CDW problem can be solved exactly and the solution explicitly written, whatever the electronic density of the chain. Owing to its theoretical importance, we discuss and demonstrate the solution anew, partly in our own way, in section 3. The remarkable result obtained by Brazovskii, Dzyaloshinskii and Krichever is due to the connection which exists between the BDK model and some classical integrable models [32,33], such as the Toda lattice [34,35] and the Volterra lattice [36] (described in section 3.5). Another important point is that the BDK model appears to be a discrete version of an earlier continuous Peierls model [37,38]. The exact solution of that model also results from the integrability of nonlinear equations, the Korteweg-de Vries (KdV) equations.

The exact solution of the BDK model is such that not only the incommensurate phases are unpinned (a result that seems to strengthen the original views of Peierls and Fröhlich [1–3]), but also, surprisingly, so are the commensurate phases. In other words, the discrete nature of the lattice seems to have no effect. This may justify the use of continuous models. However, the situation is more complex and the question which naturally arises is then: what happens beyond the range of parameters for which the model is integrable? Dzyaloshinskii and Krichever have briefly discussed the issue [39]. They have recognized that when integrability is lost, pinning should generically occur in the commensurate cases. They argue by using an analogy with a model previously proposed by Aubry [40], that in the incommensurate case, two regimes should be distinguished [39]: a weak nonlinear regime where the unpinned incommensurate phase is protected by the Kolmogorov-Arnold-Moser (KAM) theorem, and a strong nonlinear regime -the stochastic regime- where many pinned metastable configurations could appear in the system. Soon after, Aubry and Le Daeron [41] have shown that a sliding-to-pinned transition occurs for incommensurate ground-states by varying the parameters in two other discrete models: the Sue-Schrieffer-Heeger (SSH) model [42] and the Holstein model [43]. These two models are intimately related to the BDK model and appear as some limit cases (see section 2). The above sliding-to-pinned transition in the incommensurate CDW ground-state is an Aubry transition, originally called the transition by breaking of analyticity [40]. It modifies the physics of CDW systems [44]. In particular, CDWs can hardly be described in the stochastic regime by continuous models, owing to intrinsic pinning and the presence of many metastable states separated by local energy barriers [45]. Thanks to the similarity between the problem of CDW energy minimization and the time evolution of a discrete dynamical system introduced by Aubry [40], the pinning is associated with the destruction of the KAM tori of the dynamical system, when the nonlinearities become strong enough. The

concomitant apparition of chaos or stochasticity is qualitatively discussed by Dzyaloshinskii and Krichever when integrability is lost [39].

The present paper next provides a detailed numerical study of the effect of adding perturbations that either break or conserve the integrability of the BDK model. In the commensurate case (section 4.2), we establish a phase diagram when integrability is preserved, showing the relative energies of the exact solutions. When integrability is lost, we show that the CDW is always pinned, as expected on general grounds, and its shape modified. In the incommensurate case (section 4.3), we find that integrability-breaking terms trigger an Aubry transition at a certain threshold, beyond which the phase is pinned. Pinned phases are computed and have locally ordered atomic structures, which are dimers or oligomers, distinct from the exact solutions in integrable cases. The transition between pinned and sliding phases is found not only by breaking integrability, but also by changing the external pressure, making experimental verification possible.

## 2 The self-consistent electron-lattice BDK models

### 2.1 Definition

The BDK model is a combination and a generalization of two paradigmatic tight-binding models. The first one is due to Holstein and was called the molecular-crystal model [43], whereas the second is known as SSH model [42]. They are sketched in Fig. 1. The Holstein model in one dimension describes a chain of molecules $i$, each carrying an internal vibrational degree of freedom $b_i$ which modulates the on-site electronic energy level. The electrons may move along the chain with a constant hopping integral between molecules. The SSH model describes an atomic chain and was originally introduced to explain the physical properties of polyacetylene. Contrary to the Holstein model, in the SSH model there is a fixed on-site potential energy for the electrons, but the hopping integrals $a_i$ for the electrons to hop from atom $i$ to atom $i+1$ are linearly modulated by the interatomic distances $\ell_i$,

$$a_i = 1 + \alpha(\ell_i - \bar{\ell}), \tag{3}$$

where $\bar{\ell}$ is the mean lattice spacing, and $\alpha$ a parameter which can be interpreted as an electron-phonon coupling constant. The Hamiltonian of both models may be written with dimensionless variables and an overall energy scale $t$, as follows:

$$H_{\text{Holstein}}/t = -\sum_{i,\sigma}(c^\dagger_{i+1,\sigma}c_{i,\sigma} + \text{h.c.}) + \sum_{i,\sigma} b_i c^\dagger_{i,\sigma}c_{i,\sigma} + \xi\sum_i b_i{}^2, \tag{4}$$

$$H_{\text{SSH}}/t = -\sum_{i,\sigma} a_i(c^\dagger_{i+1,\sigma}c_{i,\sigma} + \text{h.c.}) + \kappa\sum_i(\ell_i - \bar{\ell})^2. \tag{5}$$

In these expressions, $i$ label the sites (atoms or molecules), $\sigma$ the electron spin, and $(c^\dagger_{i,\sigma}, c_{i,\sigma})$ are the usual (creation/annihilation) fermionic operators. $\xi$ and $\kappa$ are dimensionless parameters associated to the elastic energies of the distortions. For both models, the kinetic energy of atoms is neglected and the variables $\{\ell_i\}$ (or $\{a_i\}$) and $\{b_i\}$ are classical ones which have to be determined self-consistently, in order to minimize the energy. The spirit of these models is to keep the first linear and quadratic terms as an expansion in $b_i$ or $(\ell_i - \bar{\ell})$.



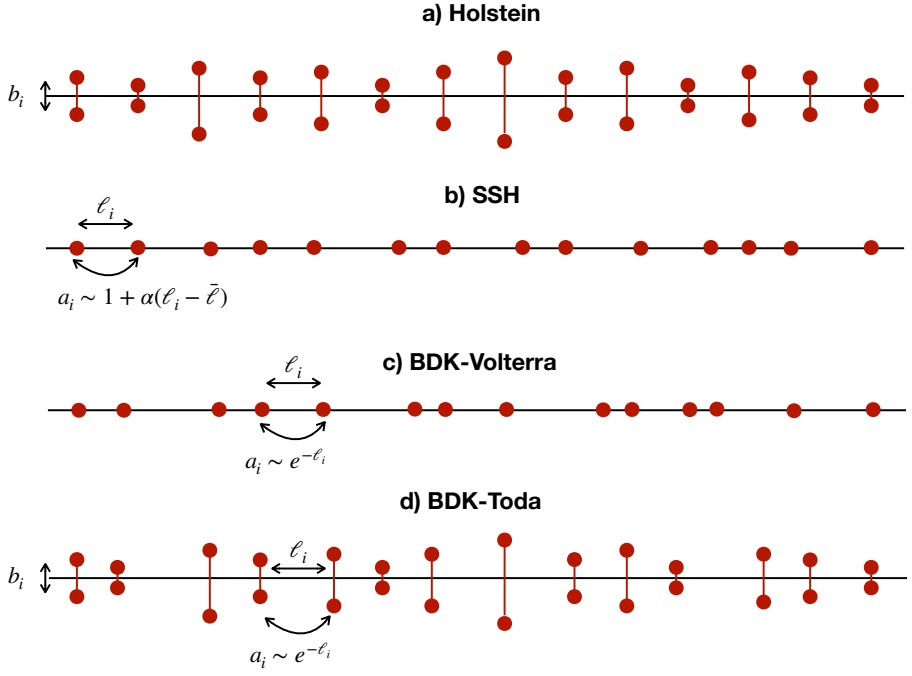

Figure 1: Illustration of the different models discussed in the text.

The BDK model [31] aims at describing the same physics of 1D chains but introduces some important differences. It similarly has two versions that distinguish the "Volterra" case from the "Toda" case.[1]

- The first BDK Hamiltonian (Volterra case) writes

$$H/t = -\sum_{i,\sigma} \frac{1}{2} e^{-\frac{\ell_i}{2}} (c^\dagger_{i+1,\sigma} c_{i,\sigma} + \text{h.c.}) + \kappa \sum_i e^{-\frac{\lambda \ell_i}{2}} + p \sum_i \ell_i \,. \tag{6}$$

It is a modification of the SSH model (see Fig. 1), in that it assumes that the hopping parameter is exponentially decreasing with the interatomic distance,

$$a_i = \frac{1}{2} e^{-\frac{\ell_i}{2}} \,. \tag{7}$$

By definition of the tight-binding approximation [6], the $a_i$ are indeed the overlap between two successive atomic (or molecular) wave functions and an exponential dependence is a fairly realistic choice. Its special dimensionless form (7) is taken for the sake of commodity, in particular to match the standard definitions in the classical integrable models. The first term in the Hamiltonian leads to an effective attraction between atoms. The second term is necessary to describe the formation of a chemical bond (see below), it is repulsive and exponentially decreasing. Its range is controlled by the parameter $\lambda$. The last and third term involves the pressure $p$ which controls the mean distance $\bar{\ell}$ between the atoms. Since the distance between sites $i$ and $i+1$ always satisfies $0 < \ell_i < \infty$,

---

[1]The use of the names "Volterra" and "Toda" is due to the connection with the classical integrable models with the same names, that will be explicit in section 3.5. Note that in the BDK paper [31], the Volterra case is also called model I, and the Toda case, model II.

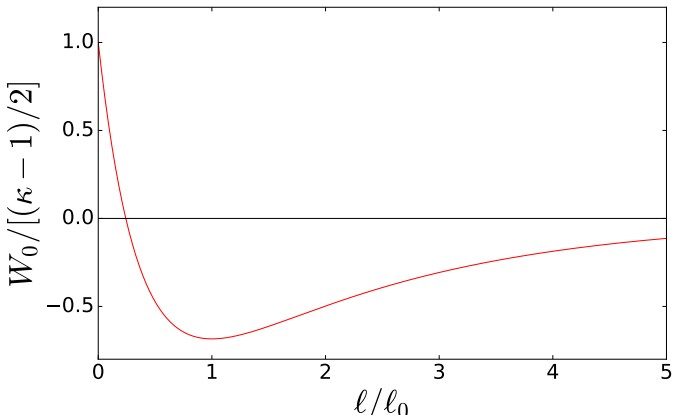

Figure 2: Two-body energy (Eq. 10). It has a minimum at $\ell = \ell_0$, the equilibrium length of an isolated covalent bond.

the variables $a_i$ satisfy the following constraint for all $i$,

$$0 < a_i < \frac{1}{2}. \tag{8}$$

The model depends on bond length variables $\{\ell_i\}$ (or $\{a_i\}$) that have to be self-consistently determined. An important difference with SSH is that it does not linearize the hopping integrals around a hypothetical metallic state. As a consequence, the issue concerning the cohesion of the chain, *i.e.* the formation of chemical bonds, is treated on an equal footing with that concerning the CDW.

• The second BDK Hamiltonian (Toda case) writes

$$H/t = -\sum_{i,\sigma}\frac{1}{2}e^{-\frac{\ell_i}{2}}(c^\dagger_{i+1,\sigma}c_{i,\sigma}+\text{h.c.})+\kappa\sum_i e^{-\frac{\lambda\ell_i}{2}}+p\sum_i \ell_i+\sum_{i,\sigma}b_i c^\dagger_{i,\sigma}c_{i,\sigma}+\xi\sum_i {b_i}^2. \tag{9}$$

It is the same as the first BDK model but it contains two additional terms which involve additional classical variables $\{b_i\}$ that describe the local Holstein vibrational degrees of freedom (see Fig. 1). They also have to be self-consistently determined.

Let us consider a first and simple example of chemical bond interactione, that of Hamiltonian (6) with two sites, two electrons and $p = 0$. The hopping of electrons between the two sites at distance $\ell$ gives a bonding state of energy $-\frac{1}{2}e^{-\frac{\ell}{2}}$, which is filled by the two electrons (spin up and spin down). The electronic energy is $-e^{-\frac{\ell}{2}}$ and the elastic cost $\kappa e^{-\frac{\lambda\ell}{2}}$. The total energy written in units of $2t$ is

$$W_0 = -\frac{1}{2}e^{-\frac{\ell}{2}} + \frac{\kappa}{2}e^{-\frac{\lambda\ell}{2}}, \tag{10}$$

which is represented in Fig. 2, after normalization. It has a minimum at

$$\ell_0 = \frac{2}{\lambda - 1}\ln(\kappa\lambda), \tag{11}$$

which is the equilibrium length of the covalent bond. The dimer is stable only when $\kappa > 1/\lambda$ and $\lambda > 1$. We will always consider that $\lambda$ and $\kappa$ satisfy these two conditions.

In the following, we are particularly interested in periodic configurations of the lattice. We call $N$ this period which is an integer, so that

$$a_{i+N} = a_i \Leftrightarrow \ell_{i+N} = \ell_i \,, \tag{12}$$

$$b_{i+N} = b_i \,, \tag{13}$$

for all $i$. The total number of sites is $L = NS$, where $S$ is the number of unit-cells in the system. We thus have $N$ independent variables to determine in the Volterra case, $\{a_1, a_2, \dots, a_N\}$, and $2N$ in the Toda case by addition of the set $\{b_1, b_2, \dots, b_N\}$.

The density per site $c$ of pairs of electrons is taken as,

$$c = \frac{r}{N} \,, \tag{14}$$

where $r/N$ is an irreducible fraction.

When the configuration is incommensurate, the period $N$ is infinite and $c$ is an irrational number. In numerical computations, it is convenient to approximate $c$ with a sequence of irreducible rational approximants $r_n/N_n$ where

$$\frac{r_n}{N_n} \to c \,, \tag{15}$$

when $n \to +\infty$. An incommensurate configuration is thus approximated by commensurate periodic configurations with period $N = N_n$.

## 2.2 General electronic band structure for an arbitrary 1D periodic lattice

We now determine the electronic band structure of the BDK models with arbitrary nearest-neighbor hoppings $\{a_i\}$ and local potentials $\{b_i\}$.

For a system with a discrete lattice periodicity $N$, the wavevectors $k$ can be chosen in the first reduced Brillouin zone $[-\frac{\pi}{N}, \frac{\pi}{N}]$. A Bloch wave function with wavevector $k$ may then be written as

$$|\Psi(k)\rangle = \sum_{i=1}^{L} \psi_i(k) c_{i,\sigma}^{\dagger} |0\rangle \,, \qquad \psi_{i+N}(k) = e^{ikN} \psi_i(k) \,, \tag{16}$$

where $|0\rangle$ is the empty state and $L = NS$ (recall that $L$ is the total number of sites and $S$ the number of unit cells). The periodic boundary conditions imply $\psi_{i+L}(k) = \psi_i(k)$ so that $e^{ikL} = 1$. We will choose $S$ even for the sake of simplicity so

$$k = \frac{2\pi n}{L} \,, \qquad \text{with} \quad -S/2 < n \le +S/2 \,. \tag{17}$$

Since the first $N$ amplitudes $\psi_i(k)$ are independent complex numbers, one defines a vector,

$$\psi(k) \equiv^t (\psi_1(k), \psi_2(k), \cdots, \psi_N(k)) \,,$$

which satisfies the eigenequation,

$$H(k)\psi(k) = E(k)\psi(k) \,, \tag{18}$$

where $H(k)$ is the $N \times N$ square matrix defined by

$$H(k) = \begin{pmatrix} b_1 & -a_1 & 0 & \dots & & -a_N e^{ikN} \\ -a_1 & b_2 & -a_2 & 0 & \dots & 0 \\ 0 & -a_2 & b_3 & -a_3 & \ddots & \vdots \\ \vdots & \ddots & \ddots & \ddots & \ddots & \vdots \\ 0 & \ddots & \ddots & \ddots & \ddots & -a_{N-1} \\ -a_N e^{-ikN} & 0 & \dots & & -a_{N-1} & b_N \end{pmatrix} \,, \tag{19}$$

in the Toda case for $N > 2$. In the Volterra case, $H(k)$ is the same, replacing $b_i$ by 0. Since $H(k)$ is an Hermitian matrix, it has $N$ real eigenvalues $E_\nu(k)$ labeled by $\nu = \{1, \cdots, N\}$, and $N$ eigenvectors $\psi_\nu(k)$. The $N$ energy bands $\{E_\nu(k)\}$ completely determine the electronic structure. Also note that $H(-k) = H(k)^*$, which implies $E_\nu(k) = E_\nu(-k)$.

We show in the Appendix A that the eigenvalues are given by solving the equation,

$$Q(E) = \cos kN, \tag{20}$$

where $Q(E)$ is a polynomial of degree $N$, written as

$$Q(E) = A_0 \left( E^N - I_1 E^{N-1} + \cdots - I_N \right), \tag{21}$$

where the amplitude $A_0$ respects $A_0 \equiv \frac{(-1)^N}{2C_N}$, with $C_N = \prod_{i=1}^{N} a_i$. By introducing the length of the unit-cell,

$$I_0 \equiv \sum_{i=1}^{N} \ell_i, \tag{22}$$

one gets $A_0 = (-1)^N 2^{N-1} e^{\frac{I_0}{2}}$. The other quantities $I_m$, $m = 1, \ldots, N$, that appear in $Q(E)$, are polynomials in the variables $\{a_i\}, \{b_i\}$. They can be derived explicitly at a given $N$, *e.g.* by iteration over the amplitudes of the eigenvectors (as shown in Appendix A) or by the direct calculation of the characteristic polynomial. As an example, for $N = 3$ in the Toda case, we have

$$\begin{aligned}
I_1 &= b_1 + b_2 + b_3, \\
I_2 &= a_1{}^2 + a_2{}^2 + a_3{}^2 - b_1 b_2 - b_2 b_3 - b_1 b_3, \\
I_3 &= b_1 b_2 b_3 - b_1 a_2{}^2 - b_2 a_3{}^2 - b_3 a_1{}^2.
\end{aligned} \tag{23}$$

For $N = 4$, in the Volterra case ($b_i = 0$), we have

$$\begin{aligned}
I_1 &= 0, \\
I_2 &= a_1{}^2 + a_2{}^2 + a_3{}^2 + a_4{}^2, \\
I_3 &= 0, \\
I_4 &= -a_1{}^2 a_3{}^2 - a_2{}^2 a_4{}^2.
\end{aligned} \tag{24}$$

The corresponding $Q(E)$ with $N = 4$ is given in Fig. 3 (a). Note that the coefficients $I_m$ with $m$ odd vanish. That remains true for any even value of $N$ in the Volterra case. As a consequence $Q(E) = Q(-E)$, so that if $E_\nu(k)$ is a solution of Eq. (20), then $-E_\nu(k)$ is another solution. The energy spectrum is symmetric with respect to $E = 0$. This symmetry is not true in the Toda case, however.

In the general case, for any $N$, the first terms can be immediately obtained, *e.g.*

$$I_1 = \sum_{i=1}^{N} b_i, \tag{25}$$

which is the trace of the matrix $H(k)$. It can be chosen to be zero by adding a constant to the energy. Similarly,

$$I_2 = \sum_{i=1}^{N} \left( \frac{1}{2} b_i{}^2 + a_i{}^2 \right) - \frac{1}{2} I_1^2. \tag{26}$$

The band structure equation (20) is an algebraic equation of degree $N$ that has $N$ real solutions, $E_\nu(k)$. We emphasize that the solutions $E_\nu(k)$ are thus some functions of $\{I_m\}$, the

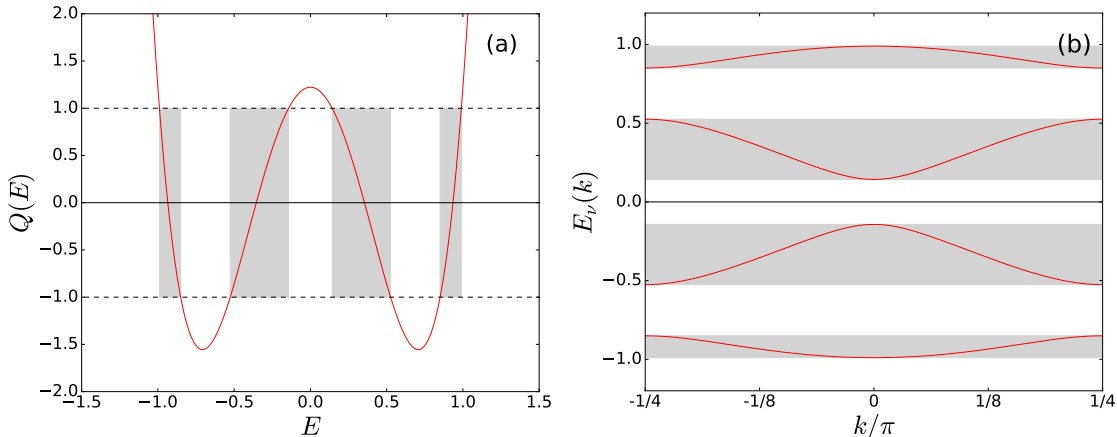

Figure 3: (a) Example of the polynomial $Q(E)$ [Eq. (21)] for a periodic problem with $N = 4$ in the Volterra case. (b) Corresponding generic band structure, $E_\nu(k)$, solution of Eq. (20). The bands are entirely determined by the three parameters $I_0, I_2, I_4$. Note the $E \to -E$ symmetry of the spectrum.

parameters of the equation, and not of the individual $\{a_i\}, \{b_i\}$ (only through the $I_m$). The band structure is thus entirely determined by a number of parameters that is smaller than the number of actual variables. Expected values of $Q$ follow

$$-1 \leq Q(E_\nu(k)) \leq 1. \tag{27}$$

The positions of the energy bands can thus be deduced from the plotting of $Q(E)$. An example with $N = 4$ is given in Fig. 3 (b): it has four bands in the reduced Brillouin zone. Note that gaps separate all the bands, which is the generic situation.

A periodic perturbation with periodicity $N$ couples the degenerate energy states labeled by $k$ and $k \pm \frac{2\pi}{N}$ and opens a gap. But the perturbation also couples the states labeled by $k$ and $k \pm \frac{4\pi}{N}, k \pm \frac{6\pi}{N}$, etc. This implies secondary gaps away from Fermi energy. However, there are specific configurations, with some symmetries of the potentials and kinetic terms, where the higher-order couplings effectively vanish, thus leaving the corresponding gaps closed.

## 2.3   Energy to be minimized for a periodic lattice

The total electronic energy $E_{elec}$ of models (6) or (9) when the chain has periodicity $N$ is obtained by filling energy bands $E_\nu(k)$ up to Fermi energy. Including spin degeneracy, it writes

$$E_{elec} = 2 \sum_{k, \nu_{occ.}} E_\nu(k). \tag{28}$$

For a concentration $c = r/N$ of pairs of electrons, the $r$ lowest bands are fully occupied: $\nu_{occ.} = 1, \ldots, r$ and the sum over $k$ extends in the whole first Brillouin zone. Thanks to the periodicity of variables $\{a_i\}, \{b_i\}$, the elastic and pressure energies are equal in all $S$ unit-cells. For example, the following sum over the $L$ sites of the chain becomes

$$\kappa \sum_i e^{-\frac{\lambda \ell_i}{2}} = S\kappa \sum_{i=1}^N e^{-\frac{\lambda \ell_i}{2}}. \tag{29}$$

It is thus convenient to define an energy $W$ per unit-cell, and use $2t$ as the unit to eliminate the spin degeneracy factor 2 in $E_{elec}$. The total energies of the two models write

- in the Volterra case:

$$W = \frac{1}{S} \sum_{k, \nu_{occ.}} E_\nu(k) + \frac{1}{2}\kappa \sum_{i=1}^{N} e^{-\frac{\lambda \ell_i}{2}} + \frac{1}{2}p \sum_{i=1}^{N} \ell_i. \tag{30}$$

- in the Toda case:

$$W = \frac{1}{S} \sum_{k, \nu_{occ.}} E_\nu(k) + \sum_{i=1}^{N} \left( \frac{1}{2}\xi b_i{}^2 + \frac{1}{2}\kappa e^{-\frac{\lambda \ell_i}{2}} \right) + \frac{1}{2}p \sum_{i=1}^{N} \ell_i. \tag{31}$$

We now consider the integrable and nonintegrable cases which differ by the values of some parameters.

### 2.3.1 Integrable BDK model

We first consider the case which was originally treated in Ref. [31], where the parameters were taken to be

$$\lambda = 2, \qquad \kappa = \xi/2^{\lambda-1}. \tag{32}$$

Equations (32) restrict the space of parameters of the model but, even if they could be only accidental, they are perfectly admissible. What matters the most is that they make the model integrable. With conditions (32), the elastic energy writes

$$\frac{1}{2}\kappa \sum_{i=1}^{N} e^{-\frac{\lambda \ell_i}{2}} = \xi \sum_{i=1}^{N} a_i{}^2 = \xi I_2, \tag{33}$$

using Eq. (26) with $b_i = 0$. Similarly, the pressure can be expressed in terms of $I_0$. Finally, one can check that the energies $W$ take the form

$$W = \frac{1}{S} \sum_{k, \nu_{occ.}} E_\nu(k) + \xi I_2 + \frac{1}{2}p I_0, \tag{34}$$

for both Volterra and Toda cases. Since $E_\nu(k)$ is a function of $\{I_m\}$, the conditions (32) implies that the total energy $W$ is only a function of $\{I_m\}$. We show later that the property $W = W(\{I_m\})$ makes the model integrable. A further generalization of BDK models that keeps this particularity has been introduced later [39]

$$W = \frac{1}{S} \sum_{k, \nu_{occ.}} E_\nu(k) + \sum_{m=0}^{N} \xi_m I_m, \tag{35}$$

for both Volterra and Toda cases, where $\xi_0 = \frac{1}{2}p$ and $\xi_2 = \xi$. For $m > 2$, $\xi_m$ are new parameters describing more general elastic energies. The expressions of $\{I_m\}$ differ between Volterra and Toda cases and are given in section 2.2. In the Volterra case, $I_m = 0$ for odd $m$.

The main feature of integrability is that, whatever the special form the energy takes, it depends on variables $\{a_i\}, \{b_i\}$ only through $I_m = I_m(\{a_i\}, \{b_i\})$. As a consequence, all solutions $\{a_i\}, \{b_i\}$ which satisfy $I_m(\{a_i\}, \{b_i\}) = C_m^{st}$ (where $C_m^{st}$ are some constants), are degenerate. We discuss this point in more details in section 3.5.

#### 2.3.2 Nonintegrable BDK model

Second, we would like to find the solutions when the integrability is broken and test which properties hold in these cases. Since we will mainly discuss the Volterra case, the simplest way to make the model nonintegrable is to choose $\lambda \neq 2$:

$$W = \frac{1}{S} \sum_{k, \nu_{occ.}} E_\nu(k) + \xi \sum_{i=1}^{N} a_i^{\ \lambda} + \frac{1}{2} p \sum_{i=1}^{N} \ell_i \,, \tag{36}$$

for Volterra case. Indeed, for $\lambda \neq 2$, the energy does not depend solely on the $I_m$. Recall that, physically, $\lambda$ is the ratio of two length scales and has no reason to be equal to 2.

For the Toda model we keep the special balance between the Jahn-Teller term and elastic repulsion, $\kappa = \xi/2^{\lambda-1}$, but allow similarly, $\lambda \neq 2$,

$$W = \frac{1}{S} \sum_{k, \nu_{occ.}} E_\nu(k) + \xi \sum_{i=1}^{N} \left( \frac{1}{2} b_i^{\ 2} + a_i^{\ \lambda} \right) + \frac{1}{2} p \sum_{i=1}^{N} \ell_i \,, \tag{37}$$

for Toda case.

The models (36) and (37) are sufficient to break integrability and are more generic. The nature of the new solutions will be addressed in section 4.

### 2.4 Nonlinear minimization equations

In order to find the chain configuration self-consistently, one needs to minimize the energy with respect to the variables of the model, for example,

$$\forall i, \quad \frac{\delta W}{\delta a_i} = 0 \,, \qquad \frac{\delta W}{\delta b_i} = 0 \,. \tag{38}$$

These are the equations for the $\{a_i\}$, $\{b_i\}$ that have to be solved. The difficulty is that they are nonlinear equations in general. Explicit equations are derived in Appendix B.

For the sake of simplicity, let us focus our study on the Volterra case. One gets

$$\frac{\delta W}{\delta \ell_i} = 0 \,, \tag{39}$$

for all sites $i$.

The linear response consists in linearizing these equations close to a non distorted uniform state, *i.e.* a metallic state of lattice constant $\bar{\ell}$. Writing

$$\ell_i = \bar{\ell} + \delta \ell_i \,, \tag{40}$$

and $\delta \ell_i \ll \bar{\ell}$, the expansion of (39) at linear order in $\delta \ell_j$ gives

$$\sum_j M_{ij} \delta \ell_j = 0 \,, \tag{41}$$

where the matrix $M$ involves the Lindhard charge or bond charge susceptibility (depending on the model which is considered) calculated for the metal (see Appendix B). If the matrix $M$ has positive eigenvalues, the only solution is $\delta \ell_i = 0$ for all $i$ and is a metal. When an eigenvalue vanishes, however, the metallic state is unstable. In a 1D chain, the Lindhard susceptibility is infinite for a modulation at $q_0 = 2k_F$, so that CDW instabilities develop at $q_0$ for infinitesimally small coupling strength.

Given that the modulation is expected at $q_0$, one assumes in the standard Peierls approach [4,5] that the solution of Eqs. (39) for the bond lengths $\ell_i$ is given by a simple cosine Ansatz,

$$\ell_i = \bar{\ell} + u \cos(q_0 i \bar{\ell} + \phi)\,, \tag{42}$$

where $u$ is the amplitude and $\phi$ a phase. For $u = 0$, the model is that of an undistorted metal with lattice parameter $\bar{\ell}$. This approach assumes a weak modulation, $u \ll \bar{\ell}$. The energy spectrum is then calculated by including the resulting perturbation, at first order in perturbation theory. The energy $W$ of the Peierls Ansatz is calculated as function of $u$ and $\phi$. Note that in the incommensurate case, the energy generally depends on $u$ but not on $\phi$: the energy as a function of $u$ and $\phi$ has a "Mexican hat" shape. The minimization of the energy with respect to $u$,

$$\frac{\delta W}{\delta u} = 0\,, \tag{43}$$

fixes the optimal modulation amplitude $u$ and the energy gap at Fermi level: it is called the "gap equation" and results from the special cosine Ansatz and from keeping the lowest order terms in $u$ in the energy including a logarithmic singularity.

In section 3, we explain the exact solutions of the BDK model at integrable points without relying on the linear response and Peierls Ansatz, which proves in general incorrect. We detail in section 4 the solutions of Eq. (39) which shape can be very different from the cosine and brings important physical consequences.

## 3 Exact solution at integrable points

In this section, we review the minimization of the total energy of BDK models, $W$, at the special points where they are integrable [Eqs. (34) or (35)]. The reader not interested in the explicit derivation may skip most of it and read the exact solutions given in section 3.6.

In integrable cases, since this energy $W$ depends only on the band structure parameters $I_m$ and if we assume that those can be varied independently, we obtain the groundstate by solving the $N + 1$ equations ($m = 0, \dots, N$),

$$\frac{\delta W}{\delta I_m} = 0\,, \tag{44}$$

which determine, in principle, the $N+1$ parameters $I_m$. This approach is different from Peierls's one which introduces the Ansatz (42) and minimizes the energy with respect to the sole parameter $u$, which, in turn, fixes the (Peierls) gap of the band structure. Equations (44) replace the standard gap equation (43).

As we will examine in details, it is not always true that the $I_m$ are independent parameters. There are some special states, which play an important role, for which it is not true. These states, called the $g$-gap states, do not have all gaps opened, as in the generic case, but only a certain number $g$. This may be seen as the consequence of some special symmetries of the chain configuration. In this case, the parameters $I_m$ are not all independent, so that the number of equations (44) changes. In fact, we will see in section 3.3.2 that the exact solution of the model in its simplest form (34), is the 1-gap Ansatz ($g = 1$) for the Toda case and the 2-gap Ansatz ($g = 2$) for the Volterra case.

In the generic case, the calculation of the gradients $\delta W/\delta I_m$ is done in section 3.1. For the $g$-gap Ansätze, for which the $I_m$ are not independent quantities, the gradients are given in section 3.2. The resulting equations are made explicit in section 3.3. The degeneracies of the solutions are discussed in section 3.4. Finally, as reviewed in section 3.5, the expressions of the $g$-gap Ansätze for the chain configuration defined by variables $\{a_i\}$, $\{b_i\}$ are explicit, thanks to a connection with classical integrable models. The solutions are summarized in section 3.6.

## 3.1 Generic candidate solution: energy gradients

Let us consider first small independent changes from $I_m$ to $I_m + \delta I_m$, keeping the period constant, and seek the corresponding linear change of $E_\nu(k)$ at a given $k$. $E_\nu(k)$, as a solution of Eq. (20), is solely a function of all $I_m$ ($m = 0, \ldots, N$), and its differential formally writes

$$\delta E_\nu(k) = \sum_{m=0}^{N} \frac{\partial E_\nu(k)}{\partial I_m} \delta I_m. \tag{45}$$

We recall that the $I_m$ themselves are functions of the variables $a_i, b_i$ and the $\delta I_m$ can be also expanded over the basis $\delta a_i, \delta b_i$ by a similar expression. From Eq. (20) and the definition (21) of $Q(E)$, we get the important relation

$$Q'(E_\nu(k))\delta E_\nu(k) - A_0 \sum_{m=1}^{N} \delta I_m E_\nu(k)^{N-m} + \frac{1}{2}\delta I_0 Q(E_\nu(k)) = 0, \tag{46}$$

for any eigenenergy $E_\nu(k)$. It can be rewritten, assuming temporarily that $Q'(E_\nu(k))) \neq 0$,

$$\delta E_\nu(k) \equiv \sum_{m=0}^{N} \frac{l_m(E_\nu(k))}{Q'(E_\nu(k))} \delta I_m, \tag{47}$$

with the definitions of the polynomials in $E$,

$$l_0(E) \equiv -\frac{1}{2}Q(E), \tag{48}$$

$$l_m(E) \equiv A_0 E^{N-m}, \qquad (m \geq 1), \tag{49}$$

which are of degree at most $N$.

The change of the electronic energy (per unit-cell) when $I_m$ are changed to $I_m + \delta I_m$ becomes,

$$\delta E_{elec} = \frac{1}{S} \sum_{k, \nu_{occ.}} \delta E_\nu(k) = \sum_{m=0}^{N} J_m \delta I_m, \tag{50}$$

with

$$J_m = \frac{1}{S} \sum_{k, \nu_{occ.}} \frac{l_m(E_\nu(k))}{Q'(E_\nu(k))}. \tag{51}$$

The corresponding variation of the total energy (35) is

$$\delta W = \delta E_{elec} + \sum_{m=0}^{N} \xi_m \delta I_m = \sum_{m=0}^{N} (J_m + \xi_m)\delta I_m, \tag{52}$$

i.e.

$$\frac{\delta W}{\delta I_m} = J_m + \xi_m, \qquad (m = 0, \ldots, N). \tag{53}$$

This is the expression of the gradient we were looking for. An explicit illustration of the derivation in the quarter-filled case, $c = 1/4$ (which will be useful in section 4.2), is given in Appendix C.

In the thermodynamic limit ($S \to +\infty$ with $N$ fixed), the sum over $k$ in Eq. (51) can be transformed into an integral. For a given density of electron pairs $c = r/N$, the electronic bands

are occupied up to $\nu = r$. Let us call $E_\nu^-$ and $E_\nu^+$, respectively the minimum and maximum of the energy for the band $\nu$. In that case, we may introduce the density of states, $\rho(E)$ and write

$$
\sum_{k,\nu_{occ.}} \to \sum_{\nu=1}^{r} \int_{-k_F}^{k_F} \frac{L\,dk}{2\pi} = \sum_{\nu=1}^{r} \int_{E_\nu^-}^{E_\nu^+} dE \rho(E), \tag{54}
$$

where the factor 2 for the spin has been removed from the very first definitions. When passing onto an integral over $E$, the $\pm k$ degeneracy implies a factor 2, so that $\rho(E) = \frac{L}{\pi}\frac{dk}{dE}$, which can be calculated directly from Eq. (20),

$$
\rho(E) = (-1)^\nu \frac{S}{\pi} \frac{Q'(E)}{\sqrt{R_{2N}(E)}}, \qquad E_\nu^- \le E \le E_\nu^+, \tag{55}
$$

$$
R_{2N}(E) \equiv 1 - Q^2(E), \tag{56}
$$

where the sign $(-1)^\nu$ makes sure that the density of states is positive in each band. $R_{2N}(E)$ is a polynomial of degree $2N$ that can be factorized: each of the root of $Q(E) \pm 1 = 0$ is a band edge (see Fig. 3), so that $R_{2N}(E) \ge 0$ and $(-1)^\nu Q'(E) > 0$ in the allowed energy bands. One obtains for Eq. (51)

$$
J_m = \sum_{\nu=1}^{r} (-1)^\nu \int_{E_\nu^-}^{E_\nu^+} \frac{dE}{\pi} \frac{l_m(E)}{\sqrt{R_{2N}(E)}}. \tag{57}
$$

Since the numerator is a polynomial and the denominator is the square-root of a polynomial, it is called a hyper-elliptic integral when $N > 2$. We will omit, thereafter, to explicitly write the limits of the integrals and the sum over $\nu$, and use the symbolic notation

$$
\sum_{\nu=1}^{r} (-1)^\nu \int_{E_\nu^-}^{E_\nu^+} \frac{dE}{\pi} \equiv \int dE. \tag{58}
$$

Either written in the form of an integral, or of a discrete sum over $k$, $J_m$ should be viewed as a function of all the band parameters $\{I_n\}$.

## 3.2 Special $g$-gap Ansätze

In general, a periodic Hamiltonian system with period $N$ has $N$ distinct bands separated by $N-1$ gaps. Some degeneracies may happen and some of the gaps may be closed accidentally or as the consequence of some symmetry of the chain configuration. Suppose that some gaps in the spectrum are closed. We are facing now an "inverse" problem: what is (are) the possible chain configuration(s), i.e. the values of the local potentials $\{b_i\}$, and hopping parameters $\{a_i\}$, that give rise to a particular spectrum, for which those gaps close (for a more general definition, see Ref. [47])?

### 3.2.1 Closing gaps, dependency relations and symmetries

In the present 1D problem, the band gaps may occur at $k = 0$ or at the Brillouin zone edge, $k = \pm\pi/N$. Suppose that two consecutive bands $\nu$ and $\nu + 1$ touch,

$$
E_\nu^+ = E_{\nu+1}^- \equiv e, \tag{59}
$$

so that there is no gap there. If the degeneracy occurs at $k = 0$, the algebraic equation, $Q(E) = 1$ (obtained from Eq. (20)), has two equal roots, i.e. a double root $e$. If instead, it occurs at $k = \pm\pi/N$, equation $Q(E) = -1$ has a double root. Wherever it occurs, $e$ is a double root of $R_{2N}(E) \equiv 1 - Q(E)^2 = 0$.

When an algebraic equation has a double root, its discriminant $\Delta$ vanishes [48]. A discriminant is a nonlinear function of the coefficients of the equation (here the $I_m$). Here, we want to vary the coefficients from $I_m$ to $I_m + \delta I_m$ with the constraint that the double root remains a double root, *i.e.* we have the constraint $\delta \Delta = 0$, which implies

$$\delta \Delta = \sum_{i=1}^{N} \left( \frac{\partial \Delta}{\partial I_m} \right) \delta I_m = 0 \,, \tag{60}$$

and gives a linear dependency relation between all $\delta I_m$. As an example, let us consider an equation of degree 2, $E^2 - I_1 E - I_2 = 0$. The discriminant $\Delta = I_1^2 + 4 I_2$ gives, at linear order, a dependency relation $\delta \Delta = 0$ which writes $2 I_1 \delta I_1 + 4 \delta I_2 = 0$. This condition ensures that, however the coefficients $I_m$ vary, degenerate solutions remain so. More generally, one may find such dependency relations at linear order in $\delta I_m$, without computing the discriminants. In our case, since a double root $e$ of equation $Q(E) = 1$ is also a simple root of $Q'(E) = 0$, the first term of Eq. (46) vanishes, $Q'(e) = 0$, and, keeping the notations defined above, one gets

$$\sum_{m=0}^{N} l_m(e) \delta I_m = 0 \,, \tag{61}$$

which is a dependency relation of the $\delta I_m$. One finds such an equation for any double root $e$ of $R_{2N}(E)$.

Equation (61) should be viewed as an Ansatz for the chain configuration, when a gap is closed. To see this more clearly, let us rewrite it as a system of equations in $a_i$ variables,

$$\sum_{m=0}^{N} l_m(e) \frac{\delta I_m}{\delta a_i} = 0 \,, \tag{62}$$

for all sites $i$. Since $I_m$ are functions of $\{a_i\}, \{b_i\}$, these equations are local equations that determine some constraints or symmetries on the hopping parameters, $\{a_i\}$ and the potentials, $\{b_i\}$. This defines some special Ansatz of the chain configuration. We give an explicit example in section 4.2.

Equations (61) are formal conditions forcing the matrices $H(k)$ to have a spectrum with a closed gap at $e$ (see also [33,49]). More generally, similar conditions apply for other problems, for example for the continuous Schrödinger equation.[2]

### 3.2.2 Energy gradients with closed gaps

When the spectrum has $g$ gaps with $g < N - 1$, the gradient of the energy, Eq. (50) can be simplified. Since the $\{\delta I_m\}$ are no longer independent in that case, the gradient can be rewritten as a sum over the independent ones. In appendix D, we prove the following result

$$\delta E_{elec} \quad = \quad \sum_{m=0}^{g+1} G_m \delta I_m \,, \tag{63}$$

which involves a sum over $g + 2$ independent terms, and where

$$G_m \quad = \quad \int dE \frac{g_m(E)}{\sqrt{P_{2g+2}(E)}} \,. \tag{64}$$

$g_m(E)$ and $P_{2g+2}(E)$ are two polynomials defined in appendix D. Note that the expression of the integrals $G_m$ also simplifies. There are still hyperelliptic in general, but with a lower degree $2g + 2$. For the solution of the BDK model, they reduce to elliptic integrals [31].

---

[2]In the case of the continuous Schrödinger equation with a periodic potential, an infinite number of gaps open in the $k^2$ spectrum, unless the periodic potential is carefully chosen. This issue is discussed in the context of periodic KdV in Ref. [49]. For example, the Schrödinger equation has a unique gap if the potential is a Weierstrass elliptic function with an amplitude precisely equal to 2 (Lamé equation).

### 3.3 Gap equations become equations for the entire band structure

#### 3.3.1 Generic case

The minimization of the energy gives

$$0 = \delta W = \sum_{m=0}^{N} (J_m + \xi_m) \delta I_m, \tag{65}$$

where $\delta W$ stands for the differential of the function $W(\{I_m\})$. For generic points in the phase space (*i.e.* almost everywhere), the $I_m$ are functionally-independent.[3] It implies the linear independence of the $\delta I_m$. Note that this is so far an assumption: there may be special solutions (at nongeneric points) where the $\delta I_m$ are dependent. This occurs solely [49] for the $g$-gap Ansätze that will be considered below. The most generic solution, if it exists, implies $J_m + \xi_m = 0$, i.e.

$$\frac{1}{S} \sum_{k,\nu_{occ.}} \frac{l_m(E_\nu(k))}{Q'(E_\nu(k))} + \xi_m = 0, \qquad m = 0, \ldots, N. \tag{66}$$

These are $N + 1$ coupled "gap equations" for the $N + 1$ unknowns $I_0, \ldots, I_N$ that determine the band structure (Toda case). In the Volterra case, only the equations with even $m$ exist. Note that the Eq. (66) are rather formal because $E_\nu(k)$ is not known in general.

One can replace the sum over $k$ by the integral (57) in the thermodynamic limit using the special notation defined in (58). For the BDK model in its simplest form ($\xi_0 = \frac{1}{2}p$, $\xi_1 = 0$, $\xi_2 = \xi$, $\xi_m = 0$ for $m \geq 3$), the "gap equations" take the following form ($N > 2$):

$$\int dE \frac{E^{N-m}}{\sqrt{R_{2N}(E)}} = 0, \quad 3 \leq m \leq N, \tag{67}$$

$$\int dE \frac{E^{N-2}}{\sqrt{R_{2N}(E)}} + \xi = 0, \quad m = 2, \tag{68}$$

$$\int dE \frac{E^{N-1}}{\sqrt{R_{2N}(E)}} = 0, \quad m = 1, \tag{69}$$

$$\int dE \frac{l_0(E)}{\sqrt{R_{2N}(E)}} + \frac{p}{2} = 0, \quad m = 0. \tag{70}$$

In these equations, starting from $m = N$, the integrals involving the polynomials $l_m(E)$ are replaced by integrals involving powers of $E$, using the fact that many of them vanish. Similarly, recalling that $l_0(E) = \frac{1}{2}Q(E)$ is a polynomial of degree $N$, but taking advantage of the fact that integrals in Eq. (67) vanish, one can replace the integral in Eq. (70) by one involving only $E^N$. Note that all these integrals are pure functions of $\{I_m\}$.

These nonlinear equations, provided they have solutions, fix the parameters $\{I_m\}$ as function of the model parameters, *i.e.* fix the entire electronic band structure and the total energy of the system. They are more general versions of the usual gap equation of CDW which controls the spectrum at the Fermi level [4].

Eventually, as shown in Ref. [31], there is no solution to this system of equations, *i.e.* there is no set of $I_0, \ldots, I_N$ that can be extracted from the model parameters $(p, \xi)$, because the integrals (67) have constant sign integrants and cannot vanish. Thus, the solution cannot be at a generic point in the phase space. For the generalized BDK model (35), however, the integrals (67) do not vanish anymore ($\xi_m \neq 0$ for $m \geq 3$) and the existence of a solution remains an open question.

---

[3]Two functions $f$ and $g$ are functionally-independent at a point $p$ if their gradients are linearly independent (noncollinear) at $p$. This is also a condition that the constants of motion in integrable systems must satisfy [51].

### 3.3.2 Special Ansätze: possible metastable states

The independence of all $\delta I_m$ assumed above is not true in all parts of the phase space. For some special values of the $a_i$, $b_i$, the $\delta I_m$ are dependent. We have seen that this is precisely the case if some gaps in the band structure are closed. When only $g$ gaps are open, one has to use Eq. (63) for the gradient to get,

$$0 = \delta W = \sum_{m=0}^{g+1} (G_m + \xi_m)\delta I_m, \tag{71}$$

which is a sum over the $g+1$ independent $\delta I_m$. Similarly,

$$G_m + \xi_m = 0, \qquad m = 0, \ldots, g+1, \tag{72}$$

giving $g+2$ nonlinear equations for the $g+2$ independent $I_m$.

- For the BDK model, Eq. (34) ($\xi_m = 0$ for $m \geq 3$), one uses Eq. (64) for $G_m$ and the definitions of the polynomials $g_m(E)$ in appendix D. As above, one can replace the integrals involving $g_m(E)$ by integrals over some powers of $E$,

$$\int dE \frac{E^{g+1-m}}{\sqrt{P_{2g+2}(E)}} = 0, \quad 3 \leq m \leq g+1, \tag{73}$$

$$\int dE \frac{g_2(E)}{\sqrt{P_{2g+2}(E)}} + \xi = 0, \quad m = 2, \tag{74}$$

$$\int dE \frac{g_1(E)}{\sqrt{P_{2g+2}(E)}} = 0, \quad m = 1, \tag{75}$$

$$\int dE \frac{g_0(E)}{\sqrt{P_{2g+2}(E)}} + \frac{p}{2} = 0, \quad m = 0, \tag{76}$$

where, for example, in the second line, the integrand contains only a term in $E^{g-1}$ since all other integrals (73) vanish. Note that these integrals exist only for $g > 1$.

If $g > 1$, following the same argument as above, there is no solution to these equations. If $g = 1$, only the last three equations remain in the Toda case, and they do have a solution in general (the integrals, which are elliptic for $g = 1$, can be inverted). We emphasize that this is the most important result obtained by Brazovskii, Dzyaloshinskii and Krichever in their paper [31]: a minimum of the energy is realized for the 1-gap Ansatz, in the Toda case. In the Volterra case, a slight modification of the argument is needed. The equations with odd $m$ (in particular Eq. (75)) do not exist and the solution is the 2-gap Ansatz [31] (see also Ref. [50]), which results from the $E \rightarrow -E$ symmetry. We present numerical check of these claims afterwards.

- For the generalized BDK model, Eq. (35), with finite $\xi_m$, the integrals (73) do not vanish anymore and other solutions with $g > 1$ may exist. All the Ansätze with different $g$ become possible solutions for the minimization equations. However, one must discard these solutions in the following cases:

  - there is no solution $\{I_m\}$ to these equations and this $g$-gap Anzatz is not an extremum.

  - a solution $\{I_m\}$ exists, but the corresponding $\{a_i\}$, $\{b_i\}$ do not respect physical constraints, for instance $\ell_i > 0$.

– a solution $\{I_m\}$ exists, but is not an absolute minimum, only a local minimum or maximum. In that case, one needs to compare its energy with that of the other solutions, in order to qualify it.

Therefore, for the generalized BDK model, many metastable states -the $g$-gap states- may coexist. We will consider in section 4.2 the simplest example of a quarter-filled band, and discuss the various possible Ansätze.

## 3.4 An implicit solution: definition of the degenerate manifold of states

The gap equations (of the previous section) allow to find, in principle, the actual values of $I_m$, for some given model parameters. Once all $I_m$ are known, the $a_i, b_i$ are implicitly determined by the equations

$$I_m = C_m^{st}, \qquad m = 0, \dots, N, \tag{77}$$

where $I_m$ are polynomials of $\{a_i\}, \{b_i\}$ (for examples, see Eqs. (23)-(24)). These equations define an algebraic manifold in the phase space, of dimension $2N$, spanned by the variables $\{a_i\}, \{b_i\}$. By construction, the energy $W$, which only depends on $\{I_m\}$, is constant on this manifold, so that it is a degenerate manifold.

How large is the degenerate manifold? Is it a single point (or a few isolated points), or is there a continuous degeneracy? Note that even the "generic" Ansatz in the current model has a very special property. Since all gaps are opened in that case, the $N + 1$ (resp. $\frac{N}{2} + 1$) nonzero $I_m$ are independent in the Toda case (resp. Volterra). The dimension of the space is $2N$ (resp. $N$), so that the manifold associated with this Ansatz has dimension $2N-(N+1)=N-1$ (resp. $N - (\frac{N}{2} + 1) = \frac{N}{2} - 1$), which implies a large continuous degeneracy and the existence of $N-1$ (resp. $\frac{N}{2} - 1$) zero modes (known as phasons in the CDW context). When some gaps are closed, the dimension is, in general, less. In particular, for the 1-gap and 2-gap Ansätze, we will see that there is a single zero mode. An exact parametrization can be given, thanks to the existence of an integrable model with explicit solutions. In this case, the degenerate ground-state manifold is a Liouville torus, as we will see next.

## 3.5 An explicit solution from the connection with classical integrable models

The problem has an explicit connection with classical integrable models, namely Toda or Volterra lattices.

A classical model with $2N$ dynamical variables is integrable in the sense of Arnold and Liouville if there are $N$ conserved quantities which are independent functions of the variables and in involution [32, 51]. In that case, there is a canonical transformation to a new set of action-angle variables $(I_m, \varphi_n)$ where $I_m$ are the $N$ conserved quantities ($\dot{I}_m = 0$) and $\varphi_n$ are the $N$ angle variables with a simple time evolution, $\varphi_n = \omega_n(\{I_m\})t$.

It is known since the 1970's that the Toda and Volterra lattices are classical integrable models and some analytic solutions of the nonlinear dynamical equations are known. From this connection, as we recall below, it is known how to construct all matrices of the form (19) (including the case when $b_i = 0$) that have a spectrum with a given number of gaps, *i.e.* it is possible to construct the $g$-gap potentials. They have been extensively studied in the past and many examples are known, *e.g.* KdV [32, 33].

### 3.5.1 Integrable Toda chain

The Toda lattice is a 1D chain of $N$ classical anharmonic oscillators governed by the following classical Hamiltonian [35],

$$H(\{p_i\}, \{x_i\}) = \sum_{i=1}^{N} \frac{p_i^2}{2} + e^{-(x_{i+1}-x_i)}, \tag{78}$$

where $x_{i+1} - x_i \equiv \ell_i$ is the dimensionless distance between two consecutive atoms at positions $x_i$ and $x_{i+1}$. By noting

$$b_i \equiv \frac{p_i}{2}, \qquad a_i \equiv \frac{1}{2} e^{-\frac{x_{i+1}-x_i}{2}}, \tag{79}$$

the nonlinear Toda equations of motion are [35, 52],

$$\begin{aligned} \dot{b}_i &= 2(a_{i-1}^2 - a_i^2), \\ \dot{a}_i &= a_i(b_i - b_{i+1}), \end{aligned} \tag{80}$$

for $i = 1, \ldots, N$. Periodic boundary conditions are assumed, *i.e.* $a_{N+1} \equiv a_1$, $b_{N+1} \equiv b_1$. These equations of motion have a remarkable property, they can be rewritten as the time evolution of a pair of "Lax matrices". The way to construct them is given explicitly by Flashka [52], who defines $H(0)$ and $A$, two matrices of size $N \times N$, by

$$H(0) = \begin{pmatrix} b_1 & -a_1 & 0 & \ldots & & -a_N \\ -a_1 & b_2 & -a_2 & 0 & \ldots & 0 \\ 0 & -a_2 & b_3 & -a_3 & \ddots & \vdots \\ \vdots & \ddots & \ddots & \ddots & \ddots & \vdots \\ 0 & \ddots & \ddots & \ddots & \ddots & -a_{N-1} \\ -a_N & 0 & \ldots & & -a_{N-1} & b_N \end{pmatrix}, \tag{81}$$

$$A = \begin{pmatrix} 0 & -a_1 & 0 & \ldots & & a_N \\ a_1 & 0 & -a_2 & 0 & \ldots & 0 \\ 0 & a_2 & 0 & -a_3 & \ddots & \vdots \\ \vdots & \ddots & \ddots & \ddots & \ddots & \vdots \\ 0 & \ddots & \ddots & \ddots & \ddots & -a_{N-1} \\ -a_N & 0 & \ldots & & a_{N-1} & 0 \end{pmatrix}. \tag{82}$$

Note that $H(0)$ is real and symmetric and $A$ is real and antisymmetric. The important point is that the equations of motion (80) can be recast in the form,

$$\dot{H}(0) = [H(0), A]. \tag{83}$$

The evolution of $H(0)(t)$ subject to this equation is unitary, *i.e.* $H(0)(t)$ and $H(0)(0)$ are related by a unitary transformation. An immediate consequence is that the $N$ eigenvalues of $H(0)$ are *conserved quantities* when the variables $a_i(t), b_i(t)$ obey the Toda nonlinear equations (80). In other words, the spectrum of $H(0)$ is invariant when the variables evolve under the Toda flow. Since there are $N$ independent conserved quantities and $2N$ variables, it is a classical integrable model in the sense of Arnold and Liouville.

The connection with the BDK problem arises because $H(0)$ is precisely the matrix defining the tight-binding problem, Eq. (19), at $k = 0$. The eigenvalues of $H(0)$ being conserved, the coefficients of its characteristic polynomial are also conserved. They are precisely the

coefficients $I_m$, defined by Eq. (21), which are therefore also constants of motion of the Toda chain. The $I_m$ form a set of action variables and the level sets $I_m = C_m^{st}$ are the Liouville tori of the classical integrable model (for corresponding angle variables, see Ref. [35]).

Back to the minimization of the Peierls problem, a solution of the Toda equations of motion, parametrized by the time $t$, conserves the same $I_m$, and since the energy $W$ depends only on $I_m$, $W$ is also conserved. The configurations $a_i(t), b_i(t)$ have the same energy $W$ than the initial configuration $a_i(0), b_i(0)$. The solution of the equations of motion thus provides continuously degenerate solutions of the Peierls model parametrized by $t$.

Some solutions of the Toda equations of motion are explicitly known. In particular, if the initial conditions $a_i(0), b_i(0)$ (or the $I_m$) are such that the matrix $H(k)$ has a single gap, then any $a_i(t), b_i(t)$ define the same spectrum with a single gap. In fact, the one-gap solution is precisely the original constant profile solution, Toda's cnoidal wave [34], rewritten as,

$$
\begin{aligned}
a_i(t) = {} & \bar{a}\left(\frac{\vartheta_3(k_0 i + vt, q)\vartheta_3(k_0(i+2) + vt, q)}{\vartheta_3^2(k_0(i+1) + vt, q)}\right)^{1/2}, \\
b_i(t) = {} & \bar{v} + v\left(\frac{\vartheta_3'(k_0 i + vt, q)}{\vartheta_3(k_0 i + vt, q)} - \frac{\vartheta_3'(k_0(i+1) + vt, q)}{\vartheta_3(k_0(i+1) + vt, q)}\right),
\end{aligned}
\tag{84}
$$

with $\vartheta_3(z, q)$ the Jacobi $\vartheta$-function defined by its Fourier series (see Appendix F for details),

$$
\vartheta_3(z, q) = 1 + 2\sum_{n=1}^{+\infty} q^{n^2}\cos(2\pi n z),
\tag{85}
$$

where $z$ is a real number here (but could be a complex number). $\vartheta_3(z, q)$ is a periodic function in $z$ with period 1 and depends on a parameter $q$, called the nome, which satisfies $0 \leq q < 1$. This parameter determines the weights of the harmonics. In the limit $q \to 0$, the function is close to a cosine with small amplitude $2q$ (and at $q = 0$, it becomes a constant, 1). When $q \to 1$, on the contrary, it contains many harmonics and becomes highly distorted. Examples of plots of $\vartheta_3(z, q)$ can be found in appendix F.

At $t = 0$, the solution depends on some parameters. Apart from the speed of the center of mass, $\bar{v}$ which describes the collective translational motion of all atoms, the solution is characterized by a dimensionless wavevector $k_0$, an amplitude $\bar{a}$ and a shape/amplitude parameter $q$. First, to ensure that the variables $a_i(t)$ and $b_i(t)$ are periodic in space, $a_{i+N}(t) = a_i(t)$, etc., one needs because of the periodicity of $\vartheta_3(z, q)$,

$$
k_0 = \frac{r}{N},
\tag{86}
$$

where $r$ is an integer. The other parameters can be chosen arbitrarily, and characterize a given initial condition.

In order for Eqs. (84) to be a solution of the nonlinear equations of motion (80), the parameter $v$ has to satisfy the condition (see Ref. [53] for a correction of the initial result of Toda),

$$
v = \pm\frac{\vartheta_1(k_0, q)}{\vartheta_1'(0, q)}\bar{a},
\tag{87}
$$

which depends on the amplitude $\bar{a}$ and $\vartheta_1(z, q)$ is defined in Appendix F.

The solution has some definite values for the conserved quantities $\{I_m\}$, such as

$$
\prod_{i=1}^{N} a_i(t) = \bar{a}^N = \frac{1}{2^N}e^{-\sum_{i=1}^{N}\ell_i(t)/2},
\tag{88}
$$

which proves that the total length is conserved. One gets

$$\bar{a} = \frac{1}{2} e^{-\frac{I_0}{2N}} \, . \tag{89}$$

Furthermore, from the second Eq. (84) and the periodic boundary conditions, one finds,

$$I_1 = \sum_{i=1}^{N} b_i(t) = \bar{v} N \, , \tag{90}$$

which is the conservation law of the total momentum or that of the trace of the matrix $H(0)$, since all eigenvalues are conserved. $I_2$ can be computed by

$$I_2 = \sum_{i=1}^{N} a_i(t)^2 - \sum_{i=1}^{N} b_i(t) b_{i+1}(t) \, , \tag{91}$$

and is conserved. Eq. (89) determines $\bar{a}$ knowing $I_0$, Eq. (90) determines $\bar{v}$ knowing $I_1$ and finally Eq. (91) determines $q$ knowing $I_2$. The other $I_m$, $m > 2$ are no longer independent parameters for the 1-gap solution, as we have seen in section 3.2.1, so that there is a complete equivalence between $(I_0, I_1, I_2)$ and $(\bar{a}, \bar{v}, q)$, the parameters of the solution. We emphasize that this is the very special cnoidal wave solution of the Toda equations of motion, which moves without deformation and can be seen as a "soliton train" [34,35]. It has a single gap in the spectrum ($H(0)$ has many doubly degenerate eigenvalues), the position of which depends on $k_0$.[4] Some other, multi-gap, solutions are known, and can be expressed with multi-dimensional $\vartheta$-functions.[5] They are more complicated and depend on further parameters. They will not be useful in the following.

It is remarkable that all matrices $H(0)$ with a single gap are known and take the form of Eq. (81) with $\{a_i\}, \{b_i\}$ explicitly given by Eqs. (84), where $vt$, noted $\phi$ below gives a free parameter for this family of solutions.

Since the exact solution of the BDK model at integrable points (in its nongeneralized version) is given by a 1-gap Ansatz, its explicit form is necessarily given by Eqs. (84). As we have seen, its explicit parameters $(\bar{a}, \bar{v}, q)$ are in direct bijection with $(I_0, I_1, I_2)$, extracted by solving the "gap equations". Therefore, the Toda-BDK model gives a continuous family of degenerate extrema, parametrized by the continuous parameter $t$ (or $\phi = vt$).

### 3.5.2 Integrable Volterra chain

Volterra nonlinear equations write [36],

$$\dot{a}_i = a_i(a_{i-1}^2 - a_{i+1}^2) \, , \tag{92}$$

for $i = 1, \dots, N$. Periodic boundary conditions $a_{N+1} \equiv a_1$ are chosen. This model is also integrable and relates to Toda's one [36]: the pair of Lax matrices involves the same matrix $H(0)$ (Eq. (81)), but with $b_i = 0$, and a matrix $A$ defined by,

$$A = \begin{pmatrix} 0 & 0 & a_1 a_2 & 0 & \dots & -a_{N-1} a_N & 0 \\ 0 & 0 & 0 & a_2 a_3 & \dots & & \\ -a_1 a_2 & 0 & 0 & 0 & \dots & & \\ \dots & & & & & & \\ 0 & \dots & & & 0 & 0 & \end{pmatrix} \, . \tag{93}$$

---

[4]Note that a potential with wave-vector $k_0$ mixes, in $k$-space, the modes $k$ and $k \pm k_0$, $k \pm 2k_0$, etc. and opens gaps at those locations in the energy spectrum. For the 1-gap potential, only the first gap is opened. When the periodicity is $N$ and the spectrum folded in the reduced Brillouin zone, the position of the gap thus depends on $r$ in $k_0 = r/N$.

[5]For the original derivation in the context of KdV, see Ref. [54]. In the context of the Toda chain, see Ref. [55].

The pair of Lax evolves again according to

$$\dot{H}(0) = [H(0), A], \tag{94}$$

when the variables $a_i$ obey Volterra equations. Thus, similarly to the previous section, Eq. (92) defines an isospectral evolution: the eigenvalues of $H(0)$ and all coefficients $I_m$ are independent of $t$. In particular, the energy defined by the BDK model (34), which is a pure function of $I_m$, does not vary in time. Any solution of (92), evolving with time, remains at the same energy and defines a degenerate manifold of states for the BDK model.

Consider the following particular solution, which is related to Toda's solution by a Bäcklund transformation [34],

$$a_i(t) = \bar{a} \left( \frac{\vartheta_3(k_0 i + vt, q)\vartheta_3(k_0(i+3) + vt, q)}{\vartheta_3(k_0(i+1) + vt, q)\vartheta_3(k_0(i+2) + vt, q)} \right)^{1/2}, \tag{95}$$

where

$$v = \pm 2 \frac{\vartheta_1(2k_0, q)}{\vartheta_1'(0, q)} \bar{a}^2. \tag{96}$$

Fixing the amplitude $\bar{a}$, the wavevector $k_0 = \frac{r}{N}$, where $r$ is an integer to ensure the periodicity, and the initial shape parameter $q$, completely determines the initial condition that will propagate as a wave, without deformation, at speed $v$. Note that the change of variables, $a_i = \frac{1}{2} e^{-\ell_i/2}$ leads to the conserved averaged length $\bar{\ell}$ and $\bar{a} = \frac{1}{2} e^{-\frac{\bar{\ell}}{2}}$. For this solution, the spectrum of $H(0)$ has two gaps (as a consequence of the symmetry with respect to $E = 0$): it is the 2-gap Ansatz which is thus the explicit solution of the BDK model in the Volterra case. Depending on $r$ in $k_0$,[4] the degenerate eigenvalues of $H(0)$ occur in a different order. As in Toda's case, other special solutions are known but will not be useful here.

## 3.6 Conclusion

In conclusion of this section, the simplest BDK model at integrable points given by Eq. (34) has explicit solutions for the ground states of the chain configuration. The ground-states form a degenerate manifold of dimension one. The expression of the ground state is parametrized by a single continuous parameter [31]:

- Toda case: the solution is the following 1-gap Ansatz. The chain configuration $\{a_i\}, \{b_i\}$ is given by Eqs. (84) (replacing $vt$ by $\phi$ and fixing $k_0 = c$):

$$
\begin{aligned}
a_i(\phi) &= \bar{a} \left( \frac{\vartheta_3(ic + \phi, q)\vartheta_3((i+2)c + \phi, q)}{\vartheta_3^2((i+1)c + \phi, q)} \right)^{1/2}, \\
b_i(\phi) &= \bar{v} + v \left( \frac{\vartheta_3'(ic + \phi, q)}{\vartheta_3(ic + \phi, q)} - \frac{\vartheta_3'((i+1)c + \phi, q)}{\vartheta_3((i+1)c + \phi, q)} \right).
\end{aligned} \tag{97}
$$

- Volterra case: the solution is the following 2-gap Ansatz. The chain configuration $\{a_i\}$ is given by Eq. (95) (replacing $vt$ by $\phi$ and fixing $k_0 = c$):

$$a_i(\phi) = \bar{a} \left( \frac{\vartheta_3(ic + \phi, q)\vartheta_3((i+3)c + \phi, q)}{\vartheta_3((i+1)c + \phi, q)\vartheta_3((i+2)c + \phi, q)} \right)^{1/2}, \tag{98}$$

$$\ell_i(\phi) = \bar{\ell} + \ln \frac{\vartheta_3((i+1)c + \phi, q)\vartheta_3((i+2)c + \phi, q)}{\vartheta_3(ic + \phi, q)\vartheta_3((i+3)c + \phi, q)}. \tag{99}$$



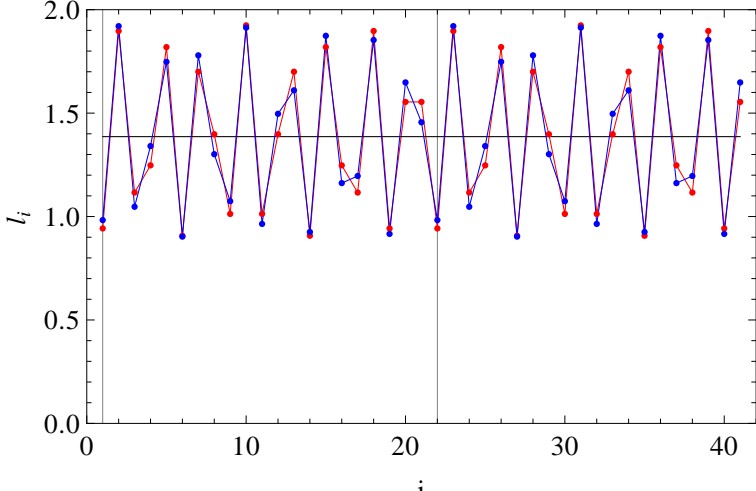

Figure 4: Example of two degenerate CDW ground states: the bond lengths $\ell_i$ according to Eq. (99) with $\phi = 0$ and $\phi = 0.03$. The period is $N = 21$ sites and 2 unit-cells are shown with a density of electron pairs $c = 8/21$, $\bar{a} = 0.25$, $q = 0.1$. The small zero energy deformation between the two curves is the phason mode.

In both cases, the energy of the configuration does not depend on the free parameter $\phi$. It ensures that the CDW can slide between the different degenerate states at no energy cost. This result, which holds at integrable points, is valid for both the commensurate and incommensurate CDW. An example of plot of $\ell_i$, Eq. (99) is given in Fig. 4 for two degenerate solutions (corresponding to two different values of $\phi$).

The parameters of the solutions $(\bar{a}, \bar{v}, q)$, or the independent parameters $\{I_m\}$ are obtained by solving the nonlinear "gap equations". The parameters $\{I_m\}$ also determine the entire band structure: the Peierls gap at the Fermi level is open (and so is the symmetric gap with respect to $E = 0$ in the Volterra case) while all other secondary gaps are closed, which is a specificity of these 1 and 2-gap Ansätze. Strictly speaking, the solutions are not only extrema, but from an analysis of small fluctuations, they are minima [56].

For the generalized BDK model at integrable points (35), all $g$-gap Ansätze are possible extrema: it is not known in this case which of them is the ground state and it certainly depends on the model parameters.

We emphasize that the degeneracies result, at least in the commensurate case, from the very special choice of the BDK model $W$. The integrability of the Toda chain, for example, ensures that there is a canonical transformation from $\{a_i\}, \{b_i\}$ to action-angle variables $\{I_m\}, \{\varphi_m\}$. A canonical transformation is a bijective nonlinear change of coordinates [51], so that *any* generic model, not necessarily at integrable points, can be expressed in terms of the new variables by

$$W(\{a_i\}, \{b_i\}) = \Omega(\{I_m\}, \{\varphi_m\}). \tag{100}$$

The BDK approach assumes that the model depends only on the $\{I_m\}$ and does not take into account the dependence in the $\{\varphi_m\}$, which would lift the degeneracy of the solutions. The motion on a Liouville torus generated by the Toda or Volterra equations implies no energy change in the Peierls model, thanks to this special choice.[6] We will illustrate the lifting of the degeneracy for a generic model in section 4.

---

[6]Alternatively, one could construct a model with an exact energy for the 1-gap Ansatz by including conjugate variables. For this Ansatz, all but one angular variables are time-independent [35], so that if $W$ depends on the unique time-dependent variable $\varphi$, the degeneracy is lifted: the solution describes a pinned CDW with a Peierls-Nabarro barrier.

# 4 Beyond integrability

In this section, our concern is to understand what happens when the BDK model departs from integrable points in the parameter space. We will focus exclusively on the Volterra case, but our qualitative conclusions remain the same in the Toda case.

On general grounds, one expects that the degeneracy of the exact solutions, parametrized by the continuous parameter $\phi$ that describes free sliding, should be lifted, because the BDK model would no longer depend solely on the coefficients $\{I_m\}$ (see Eq. (100)). In that case, the states would be pinned. The situation is, however, more subtle. The change of regime from freely sliding states to pinned states is characteristic of an Aubry transition and we will show that such transitions occur here.

The Aubry transition, originally called the transition by breaking of analyticity, is associated to nonlinear terms in the equations as well as to chaotic transitions [40, 57, 58]. Indeed, the minimization equations,

$$\forall i, \quad \frac{\delta W}{\delta \ell_i} = 0, \tag{101}$$

are essentially nonlinear equations for the chain configuration variables $\{\ell_i\}$ (for example see Eq. (B.7) in appendix B). They can be viewed as a discrete-time map (where $i$ plays the role of discrete time) where $\ell_i$ is a function of the previous $\ell_j$, $j < i$. Since this map is nonlinear, a chaotic transition may take place when the nonlinearity is strong enough: from the regular integrable solutions, the system can enter a stochastic regime characterized by the destruction of KAM tori and the rising of chaotic regions in the phase space [40, 57, 58]. This is the stochastic regime qualitatively suggested by Dzyaloshinskii and Krichever [39] for the present model. On the other hand, when the nonlinearity is small enough, KAM tori (evolving from Liouville tori of the integrable system) may nevertheless survive in incommensurate cases, thus preserving integrable sliding solutions. In a one-dimensional system as it is the case here, the Aubry transition between the two regimes is characterized by the apparition of discontinuities in the envelope function of the CDW modulation (which are nonanalyticities due to the destruction of KAM tori) [41, 44], which is defined and computed below for the present model. Furthermore, in the stochastic regime, many metastable states emerge (at higher energies) due to chaotic regions and coexist with the ground-state. It is important to stress that, although the ground-states are obtained inside the stochastic (chaotic) regime, they remain either periodic (in the commensurate case) or quasi-periodic (in the incommensurate case).

After the definition of the envelope function in section 4.1, we examine two examples, a commensurate case, $c = 1/4$ (in section 4.2) and an incommensurate case, $c = (3 - \sqrt{5})/2$ (in section 4.3).

## 4.1 Definition of the envelope function, Aubry breaking of analyticity

Consider a wave with a modulated amplitude $a_i$ at site $i$ and wavevector $2\pi c$. The definition of the envelope function[7] is aimed at filtering out the fast oscillations of the wave, which are necessarily discontinuous for a model with discrete sites (such as that in Fig. 4). It will replace a fast cosine modulation, e.g. $a_i = u \cos(2\pi i c)$ by the 1-periodic function, $f(\phi) = u \cos(2\pi \phi)$, i.e. $f(ic) = a_i$. This definition allows one to address the issue of the continuity or analyticity of the envelope function $f$.

In the incommensurate case, the wave is not periodic and $c$ is an irrational number. There is an infinite number of distinct amplitudes $\{a_i\}$. The envelope function $f$ is defined to be a

---

[7]It is also called the "hull" function. The envelope of a wave is generally defined as a smooth curve connecting the maxima or minima, this is not the definition we use here.

periodic function with period 1, *i.e.* $f(\phi + 1) = f(\phi)$, and

$$f(ic) = a_i. \tag{102}$$

Since the numbers $ic \pmod 1$ cover densely the interval $[0,1]$ when $c$ is irrational, the function $f$ is well-defined.

In the commensurate case, the wave is periodic with $c = r/N$ and period $N$. The wave is entirely determined by a finite number of amplitudes $\{a_1, \ldots, a_N\}$. The numbers $\phi_i = ic \pmod 1$ take $N$ discrete values inside $[0,1]$, so that the function

$$f(\phi_i) = a_i, \tag{103}$$

is defined only at the $N$ discrete points $\phi_i$. To extend $f$ to the entire interval $[0,1]$, one could choose other close commensurabilities, for example,

$$c_m = \frac{mr + r_{p-1}}{mN + N_{p-1}}. \tag{104}$$

Indeed, when $m$ increases, $c_m \to c = r/N$ and the points $\phi_i = ic_m \pmod 1$ fill more and more the interval $[0,1]$. It thus makes sense to define an envelope function $f$ in $[0,1]$ as the limiting function when $m \to +\infty$. In practice, in the incommensurate case as well, we consider a sequence of rational approximants of $c$, $c_n = r_n/N_n$ (see Eq. 15), and only define the envelope functions $f_n(\phi_i)$ at the $N_n$ points, $\phi_i = ic \pmod 1$. When $n$ increases, the number of points $N_n$ increases and $f_n(\phi_i)$ will converge onto $f(\phi)$ defined in the interval $[0,1]$.

At integrable points (as in section 3), we have analytic solutions of the form,

$$a_i(\phi) = f(ic + \phi), \tag{105}$$

where $\phi$ characterizes a given solution and $f$ is periodic of period 1, see *e.g.* Eq. (98). If $c$ is irrational, it is thus exactly of the form (102). If $c$ is rational, there is no difficulty since $f$ does not depend on $c$ and one can choose a close irrational number for example. $f$ is thus the envelope function. At integrable points, it is, therefore, a *continuous* function, both in commensurate and incommensurate cases.

Away from the integrable regime, we will see the emergence of *discontinuities* in the envelope function: this is the breaking of analyticity characterizing the Aubry transition. We will argue that it occurs abruptly for the commensurate case. For the incommensurate case, the continuity of the envelope function may persist away from integrable points, at least in certain regions of parameters.

## 4.2 Example of commensurate solution: $c = 1/4$

In this section, we consider the BDK model (Volterra case) at quarter filling $c = 1/4$. Since the chain has a periodicity of $N = 4$ sites in this case, the present example is the simplest, yet nontrivial example where the BDK results can be tested: the chain is defined by the only four inequivalent bond length variables, $(\ell_1, \ell_2, \ell_3, \ell_4)$.

At integrable points, the ground state of the simplest BDK model [Eq. (34)] was argued to be the 2-gap Ansatz. We will compare that result with the numerical one. Furthermore, we consider two kinds of perturbations: one that conserves integrability (the generalized BDK model of Eq. (35)) in sections 4.2.1-4.2.3. This allows us to understand whether or not the other $g$-gap Ansätze with $g > 2$ could be the ground state of some generalized model. In section 4.2.4, we consider a perturbation that explicitly breaks integrability and compute the solutions numerically.

#### 4.2.1 Solving the "gap equations" at integrable points

At integrable points, the energy of the generalized BDK model takes the form

$$W = \frac{1}{S} \sum_k E_1(k) + \sum_{m=0}^{2} \xi_{2m} I_{2m}, \tag{106}$$

where only the first band (out of four) is filled. The model parameters are $\xi_0 = p/2$, $\xi_2 = \xi$ and $\xi_4$ (defined in Appendix C). The model depends solely on the $\{I_m\}$ and the minimization leads to (Eq. (65)),

$$\delta W = \sum_{m=0}^{2} (J_{2m} + \xi_{2m}) \delta I_{2m} = 0, \tag{107}$$

$J_0, J_2, J_4$ are given in Appendix C. By considering the different $g$-gap Ansätze, we obtain different "gap equations". Solving them allows to extract the optimal band structure parameters $\{I_m\}$, and determine the stability of the different phases in parameter space and the corresponding energies.

*Generic 3-gap Ansatz*

In the most generic case, the four bands are separated by three gaps, as in Fig. 3. We expect the distortions to be generic in the phase space, so that all $\delta I_m$ must be independent. From Eq. (107) and the definitions of $J_0, J_2, J_4$ in Appendix C, the "gap equations" then write

$$\frac{1}{S} \sum_k \frac{Q(E_1(k))}{Q'(E_1(k))} = p, \tag{108}$$

$$\frac{1}{S} \sum_k \frac{A_0 E_1(k)^2}{Q'(E_1(k))} = -\xi, \tag{109}$$

$$\frac{1}{S} \sum_k \frac{A_0}{Q'(E_1(k))} = \xi_4. \tag{110}$$

They are three nonlinear equations for the three unknowns $I_0, I_2, I_4$. We have solved them numerically for given model parameters. However, it is not always possible to find a set of $I_0, I_2, I_4$ satisfying these equations. For example, if $\xi_4 = 0$, there is no solution to these equations, as originally shown by BDK. Indeed, it is clear that the left-hand-side of Eq. (110) has $Q'(E_1(k)) < 0$ for all $k$ (see Fig. 3 (a)) and is therefore strictly negative: $\xi_4$ must be negative, in order to have a chance to have a 3-gap solution. Similarly it is also clear that $p$ and $\xi$ must be positive. The zone where the three equations have a solution that, furthermore, respects the constraints (C.5), is shown by the grey area in Fig. 5. A direct way to determine it is to compute the three sums for a large sample of physically-allowed coefficients $I_0, I_2, I_4$ which values are chosen in a compact set. Outside this area, the 3-gap phase is no longer an extremum and other phases have to be sought. Once $I_0, I_2, I_4$ are known, the energy $W$ of the 3-gap Ansatz (which depends only on $I_0, I_2, I_4$) can be computed. It is given as function of $\xi_4$ in Fig. 6 (green circles) for $p = 0.02$ and $\xi = 1$, as an example.

*2-gap Ansatz*

Consider the situation of Fig. 7 where the gap at $E = 0$ ($k = 0$) is closed. By imposing this extra condition, which implies a condition on the $\{I_m\}$, we impose a dependency relation of the $\delta I_m$, as we have seen in section 3.2.1. First, which symmetry of the chain configuration does correspond this degeneracy? The degeneracy implies that the discriminant of the algebraic equation $Q(E) = 1$ vanishes, *i.e.* (see appendix C),

$$I_4 = -2C_4. \tag{111}$$

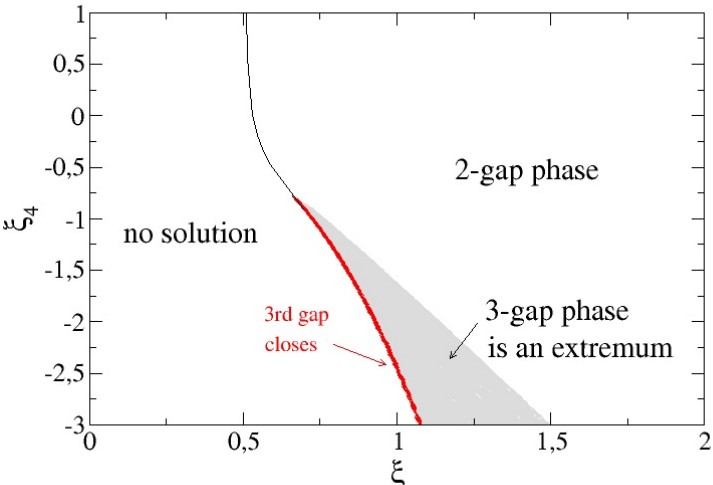

Figure 5: Regions of existence for the various phases at $c = 1/4$: the 3-gap phase exists only in the grey area. The 2-gap phase exists everywhere except in the "no solution" region where the constraints, $\ell_i > 0$ for all $i$, are not fulfilled. The 0-gap phase (not shown) exists wherever the other phases exist. Here, $p = 0.02$.

Deriving this relation with $C_4 = \frac{1}{16} e^{-I_0/2}$ leads to

$$\delta I_4 + \frac{1}{2} I_4 \delta I_0 = 0 \,, \tag{112}$$

which indicates that $\delta I_4$ and $\delta I_0$ are dependent. Note that it can be found without the explicit form of the discriminant, thanks to Eq. (46). From the definition, $I_0 = -2 \sum_i \ln 2a_i$, $\frac{\delta I_0}{\delta a_i} = -\frac{2}{a_i}$, one obtains a series of equations,

$$a_i \frac{\delta I_4}{\delta a_i} = I_4 \,, \tag{113}$$

which implies that $a_i \frac{\delta I_4}{\delta a_i}$ is a constant, independent of $i$. With the definition of $I_4$, Eq. (C.3) in Appendix C, these equations become,

$$a_1{}^2 a_3{}^2 = a_2{}^2 a_4{}^2 = -\frac{I_4}{2} \,. \tag{114}$$

It means that the distortions $u_i$, away from the average, $\ell_i \equiv \bar{\ell} + u_i$ satisfy

$$u_1 = -u_3 \quad \text{and} \quad u_2 = -u_4 \,. \tag{115}$$

This is a particular symmetry of the chain modulation. Any perturbation of the Hamiltonian respecting this symmetry, whatever its strength, leaves the gap at zero energy closed.[8] This is the special 2-gap Ansatz for $N = 4$.

In the minimization equation (107), use the dependency relation (112) to replace $\delta I_4$, and the independence of $\delta I_0$ and $\delta I_2$ to obtain the two "gap equations",

$$J_0 + \frac{p}{2} + \frac{1}{2} \xi_4 I_4 + \frac{1}{2} J_4 I_4 = 0 \,, \tag{116}$$

$$J_2 + \xi = 0 \,, \tag{117}$$

---

[8]This is somewhat different from a periodic perturbation in the continuous Schrödinger equation, where one-gap potentials have *finite-amplitudes* (see footnote 2).

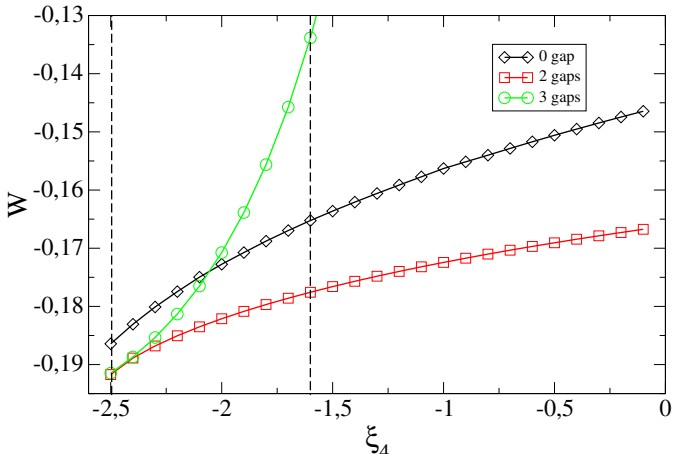

Figure 6: Comparison of energies for the various solutions, 0-gap, 2-gap, or 3-gap. We see that the 2-gap Ansatz is always at lower energy, whatever $\xi_4$. Here $p = 0.02$, $\xi = 1$. The values of $\xi_4$ between the two dashed lines correspond to the physical region of stability of the 3-gap state for $\xi = 1$ (see Fig. 5).

which are rewritten

$$\frac{1}{S} \sum_k \frac{1 - \cos kN}{Q'(E_1(k))} = -p - \xi_4 I_4 \,, \tag{118}$$

$$\frac{1}{S} \sum_k \frac{A_0 E_1(k)^2}{Q'(E_1(k))} = -\xi \,, \tag{119}$$

thanks to $A_0 I_4 = -1$ (which is zero discriminant condition) and $Q(E_1(k)) = \cos kN$. These are two nonlinear equations for two unknowns variables $I_2$ and $I_4$, which are solved numerically. Alternatively, BDK replaces the sum over $k$ by elliptic integrals that can be inverted with elliptic functions. It turns out that they have a solution in the whole range of the parameter space shown in Fig. 5: the physical constraints Eqs. (C.5) are also fulfilled, except in the region called "no solution". The energy of the 2-gap Ansatz is given in Fig. 6 (in red squares).

*0-gap Ansatz (metal)*

Consider finally that the gaps at $k/\pi = 1/4$ in Fig. 7 vanish (both the low energy and the high energy gaps have to vanish simultaneously). As above, it is possible to find an additional dependency relation from the zero discriminant. In this case, we have an undistorted metallic chain respecting a translational symmetry with a single parameter to optimize, its length. Its optimal energy is obtained in the thermodynamic limit and shown in Fig. 6 (black diamond).

By comparing the energies of the three considered states in Fig. 6, we see that the 2-gap Ansatz is always the lowest energy state, whatever $\xi_4$. It is lower than that of the metal (which proves the Peierls transition). It is also lower than the 3-gap Ansatz in the stability region (in between the two dashed lines) where both states coexist. The two energies are very close for $\xi_4 \sim -2.5$, because the third gap of the 3-gap Ansatz closes continuously and the two states merge at the same time. These results have been confirmed by a direct numerical minimization (see section 4.2.3).

All the $g$-gap solutions may be extrema of the energy, at the condition that the "gap equations" have a solution. In the present example, where a perturbation $\xi_4$ is added to the stan-

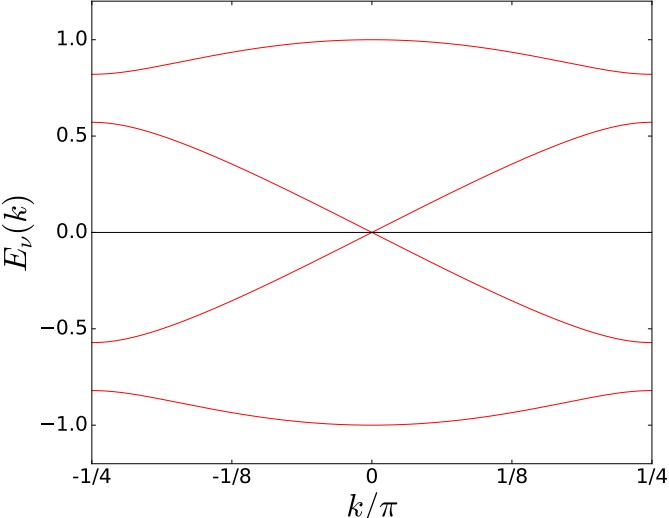

Figure 7: Band structure of the 2-gap Ansatz for $N = 4$: the bands have now two gaps. The gap at $E = 0$ is closed (compare with Fig. 3).

dard BDK model, the 2-gap Ansatz still remains the lowest energy state.

### 4.2.2 Implicit solution: degenerate manifold of states

Once the parameters $I_0$ (or $C_4$), $I_2, I_4$ are known for given model parameters (as shown in the previous section), one can obtain the distortions, $a_i$. They can be obtained by solving the equations,

$$I_2 = a_1{}^2 + a_2{}^2 + a_3{}^2 + a_4{}^2, \tag{120}$$

$$I_4 = -a_2{}^2 a_4{}^2 - a_1{}^2 a_3{}^2, \tag{121}$$

$$C_4 = a_1 a_2 a_3 a_4, \tag{122}$$

where $I_2, I_4, C_4$ are given constants. Three equations in a 4D space $(a_1, a_2, a_3, a_4)$ define, in general, a 1D manifold. All points on this curve have the same $I_m$ and are therefore degenerate: they are all equally valid solutions. It is easy to use Eqs. (121) and (122) to eliminate $a_3$ and $a_4$ and obtain two possible solutions,

$$I_2 = a_1{}^2 + a_2{}^2 - \frac{I_4}{2}\left(\frac{1}{a_1{}^2} + \frac{1}{a_2{}^2}\right) \pm \frac{\sqrt{I_4^2 - 4C_4^2}}{2}\left(\frac{1}{a_1{}^2} - \frac{1}{a_2{}^2}\right), \tag{123}$$

which defines two curves in the $(a_1, a_2)$ plane, when $I_4 \neq -2C_4$ (Fig. 8, left) or one when $I_4 = -2C_4$ (Fig. 8, right). The solution for $(a_1, a_2)$ is anywhere on these curves, and $(a_3, a_4)$ are simply obtained from

$$a_1{}^2 a_3{}^2 = \frac{-I_4 \pm \sqrt{I_4^2 - 4C_4^2}}{2}, \tag{124}$$

$$a_2{}^2 a_4{}^2 = \frac{-I_4 \mp \sqrt{I_4^2 - 4C_4^2}}{2}. \tag{125}$$

We note that $a_1 a_3 \neq a_2 a_4$ when $I_4 \neq -2C_4$ and can be interchanged by symmetry (they play the same role in the translation). When $I_4 = -2C_4$,

$$a_1{}^2 a_3{}^2 = a_2{}^2 a_4{}^2 = -\frac{I_4}{2}, \tag{126}$$

which is the symmetry responsible for the absence of zero-energy gap in the spectrum of the 2-gap state, already discussed in Eq. (114). One can define an order parameter,

$$M = a_1{}^2 a_3{}^2 - a_2{}^2 a_4{}^2 = \pm\sqrt{I_4^2 - 4C_4^2}\,, \tag{127}$$

that is zero in the 2-gap phase and nonzero in the 3-gap phase.

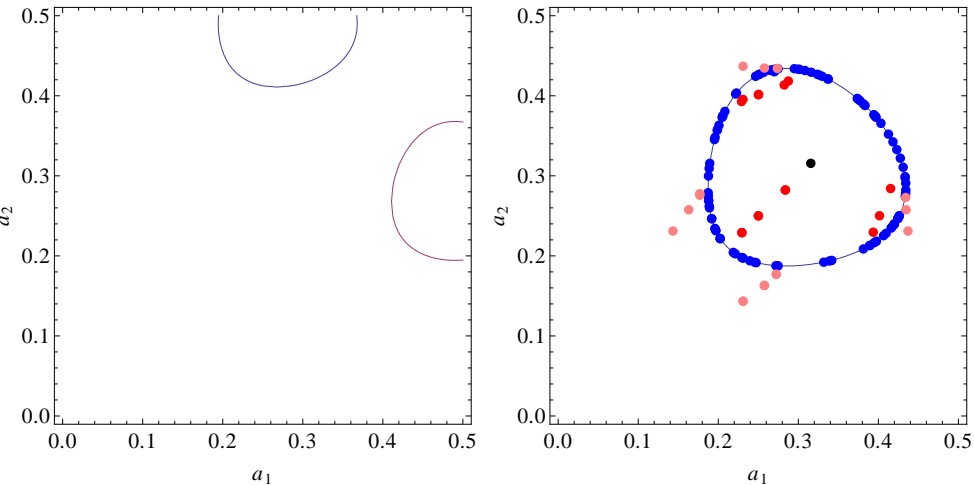

Figure 8: Degenerate 1D manifolds showing the possible values of $a_1, a_2$ for the 3-gap Ansatz (left) and the 2-gap Ansatz (right, continuous line) for $p = 0.02, \xi = 1$ and $\xi_4 = -2$ [Eq. (123)]. The discrete blue points are from direct numerical minimization starting from random configurations (section 4.2.3), and fall precisely onto the calculated line. The black point is the optimal metal (same parameters). Away from integrability ($\lambda \neq 2$), the continuous degeneracy is lifted since all the random configurations fall onto the same set of discrete points, related by translation symmetry: pink points for $\lambda = 1.98, 1.95, 1.9$, red points for $\lambda = 2.03, 2.06, 2.1$ (same other parameters).

In both cases (2 or 3 gaps), the solutions lie in a 1D manifold (with one or two curves), so there is a continuous degeneracy that can be characterized by a single zero-energy phason mode. The solutions are unpinned commensurate CDW. The metal (or 0-gap) state is obtained from the 2-gap one when the "radius" of the 1D manifold vanishes. Indeed, the right-hand-side of Eq. (123) has a minimum which is reached at the uniform state $a_1{}^4 = a_2{}^4 = -\frac{I_4}{2}$ when $I_4 = -2C_4$. This minimum is represented by a point (in black on the diagonal $a_1 = a_2$ of Fig. (8). The degenerate manifold continuously evolves when the gaps successively close, by the coalescence of the two curves onto a single one and the shrinking of the remaining curve onto a point.

In summary, for commensurate quarter filling, the distortions are nonzero but the modulation can be continuously deformed without energy cost. The sliding property results from the special choice of the energy that depends only on the $I_m$ of the band structure. For larger $N$, the degenerate manifold is of higher dimension, and one has, in general, more phason modes. For given model parameters, we have found that the lowest energy state is the 2-gap state. This approach allows one to visualize the degenerate chain configurations, even for the higher-energy Ansätze.

We now solve numerically the minimization problem for the variables $a_i$, instead of solving the nonlinear "gap equations" and taking a solution from the degenerate manifold. This allows one to check whether the lowest energy state found in section 4.2.2, is indeed a minimum.

Since one knows, from the Volterra classical integrable model, the exact form of the distortions that give a spectrum with 2 gaps, one can compare the numerical solution with the exact one. Furthermore, one can modify the model, such as adding terms that cannot be expressed in terms of the $\{I_m\}$ and observe how the commensurate phases acquire some pinning, as generally expected.

### 4.2.3 Numerical minimization compared to the exact solution

For a given set of model parameters at integrable points, we find the ground-state configurations of the chain $(a_1, a_2, a_3, a_4)$ by a direct numerical minimization of the energy $W$ (see appendix E). The various solutions are obtained by minimization of many randomly distorted chains as initial conditions: many degenerate solutions are thus obtained. For example, for $p = 0.02$, $\xi = 1$, $\xi_4 = -2$, the solutions are shown in Fig. 8 (right) in blue points: they fall perfectly onto the expected degenerate manifold of the 2-gap Ansatz (in solid line). Once the $\{a_i\}$ are known, the values of $I_m$ are computed and are in perfect agreement with those extracted by solving the "gap equations". Furthermore, we find that the 2-gap phase is always the ground state, as originally shown by BDK for $\xi_4 = 0$. When $\xi_4$ varies, the 3-gap phase is never found as a minimum, although it corresponds to a higher-energy extremum in the parameter region delimited by the grey area in Fig. 5. It may be either a maximum or a metastable state with a small basin of attraction.

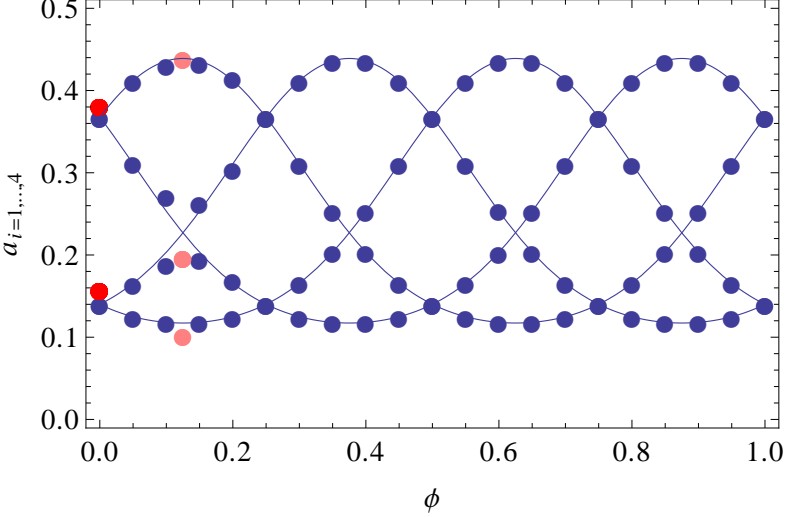

Figure 9: Various degenerate solutions for the CDW distortions $a_1, \ldots, a_4$ in the integrable case ($\lambda = 2$): numerical results (blue circles) and exact results (lines). Here $p = 0.02$, $\xi = 1$, $\xi_4 = 0 \rightarrow I_0 = 6.321$, $q = 0.225$. In the nonintegrable case ($\lambda \neq 2$), the solutions are non degenerate and are shown by the pink points ($\lambda = 1.9$) -the middle point is twice degenerate, and by the red points ($\lambda = 2.1$) -the two points are twice degenerate, corresponding to the structures shown in Fig. 11.

In Fig. 9, we plot directly the four $a_i$ for various degenerate solutions (in blue solid circles). Numerically, the points are obtained by fixing the first distortion $a_1$ to various exact values and let the other distortions relax in the minimization procedure. Fixing $a_1$ is done here for representation purpose, so as to plot as a function of a phase $\phi$. The exact result for the 2-gap Ansatz is given by Eq. (98):

$$a_i(\phi) \;=\; \bar{a}\left(\frac{\vartheta_3(ic + \phi, q)\vartheta_3((i+3)c + \phi, q)}{\vartheta_3((i+1)c + \phi, q)\vartheta_3((i+2)c + \phi, q)}\right)^{1/2}, \tag{128}$$

for $i = 1, \ldots, 4$. The parameters of the solution are all fixed (except $\phi$): $c = 1/4$ ensures that the single gap opens at the Fermi energy; $\bar{a} = \frac{1}{2}e^{-\frac{I_0}{2N}}$, $I_0$ and $I_2$ were obtained from section 4.2.1 by solving the "gap equations". Since $I_2 = \sum_{i=1}^{N} a_i{}^2$ is a function of $q$, the parameter of the $\vartheta_3$ function, $q$ is numerically determined. Therefore, in the exact solution above, there is no free parameter left, except $\phi$ which reflects the continuous degeneracy, *i.e.* the one-parameter family of ground states. The four curves $a_i(\phi)$ are plotted in Fig. 9 and are in perfect agreement with the numerical results, thus confirming that the 2-gap Ansatz is the ground state at integrable points.

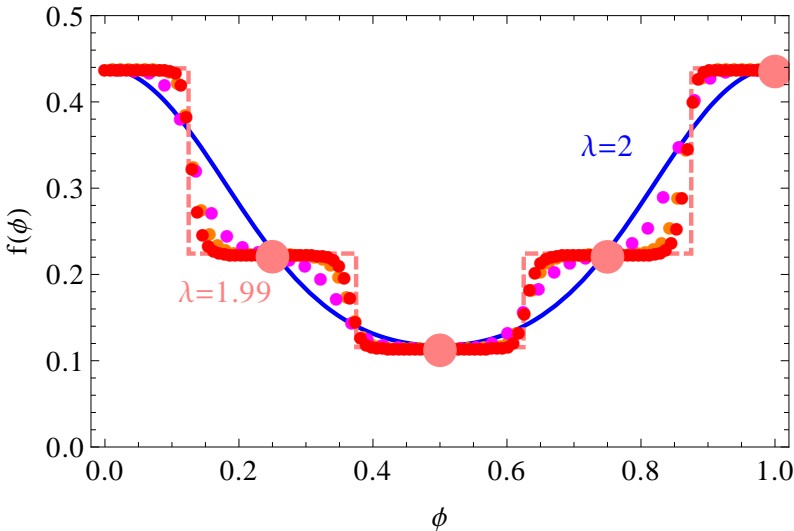

Figure 10: Envelope functions defined by Eq. (105) at the integrable point $\lambda = 2$ (continuous blue curve) and computed numerically at $\lambda = 1.99$: the points for $c = 11/43, 21/83, 31/123$ (magenta, orange, red) converge onto the limit envelope function of $c = 1/4$ (discontinuous dashed line at the limit). The four points for $c = 1/4$ at $\lambda = 1.99$ are also shown (large pink points). Here $p = 0.02$, $\xi = 1$. There is thus an abrupt Aubry transition at $\lambda = 2$ in this simple commensurate case $c = 1/4$.

Two special configurations have to be noticed. First, note that a change of $\phi$ by $\phi + 1/4$ corresponds to a translation of the bonds, so that in Fig. 9 the same translated configurations are found. At $\phi = 0$ (or equivalently at $\phi = \frac{1}{4}, \frac{1}{2}, \frac{3}{4}$), the configuration consists of two neighboring short bonds of equal length (a trimer) followed by two long bonds of equal length. At $\phi = \frac{1}{8} = 0.125$ (or at equivalent points), the configuration has a single short bond (a dimer) surrounded by two equal-length bonds, followed by a longer bond. As we have emphasized, these configurations are degenerate with many others.

The envelope function $f$, defined by Eq. (105), is thus directly obtained from Eq. (128). $f$ is periodic with period 1, and continuous in this integrable case. It is represented by the solid line (for $\lambda = 2$) in Fig. 10.

### 4.2.4 Numerical minimization for nonintegrable cases: Aubry transition

We now consider the BDK model (36) away from integrable points,

$$W = \frac{1}{S} \sum_k E_1(k) + \xi \sum_{i=1}^{N} a_i{}^{\lambda} + \frac{1}{2} p \sum_{i=1}^{N} \ell_i. \tag{129}$$

When the ratio of length scales $\lambda \neq 2$, the model does not depend solely on the $\{I_m\}$ and is no longer integrable. The minimization of $W$ is therefore now exclusively numerical: the bond lengths $\{\ell_i\}$ (or $\{a_i\}$) are self-consistently determined numerically (see Appendix E).

For $\lambda < 2$, we find the chain configuration shown in Fig. 11 (left): the four atoms are represented in their effective positions (atom 1 is fixed at the origin). The short bond (here between atoms 2 and 3) is represented by a thick line and corresponds to the formation of a dimer, where the pair of electrons ensures the covalent bonding: the unit-cell, shown by a dashed rectangle, is periodically repeated, giving a

$$L_1 S L_1 L_2 L_1 S L_1 L_2 \dots \tag{130}$$

sequence, where $L_1$ and $L_2$ are two different long bonds and $S$ is the short dimer bond. The electronic density is stronger on the two sites of the dimer, but nonzero elsewhere and is shown in Fig. 11 by the color code. Contrary to the integrable case, starting the minimization from many random configurations always leads to the same solution, up to the four translations (which move the strong bond to any one of the four bonds). The continuous degeneracy has been lifted and one of the solutions has now the lowest energy: the solution corresponds to a slight modification of the solution $\phi = \frac{1}{8}$. The comparison is given in Fig. 9 where this solution is indicated by pink circles (the middle circle corresponds to the two equal distortions). Similarly, in Fig. 8, the new solution, instead of being continuously degenerate, corresponds to four discrete pink circles, which can exchanged thanks to the translational symmetry.

It is still possible to define an envelope function $f$, as explained in section 4.1. For $c = 1/4$, the numbers $ic$ (mod 1) take only four values, 1/4, 1/2, 3/4, and 1 (= 0 (mod 1)), so that $f$ can be defined only at those four points by $f(1/4) = a_1$, $f(1/2) = a_2$, $f(3/4) = a_3$ and $f(1) = a_4$. They are represented by the large pink points in Fig. 10. To define the function outside these points, consider the commensurabilities $c_m = (m+1)/(4m+3)$, in particular $11/43, 21/83, 31/123$ which are closer and closer to $c = 1/4$. The points $ic_m$ (mod 1) fill more and more densely the interval $[0, 1]$ when $m$ increases and a true envelope function can be defined by the limit of successive set of points and is represented by a dashed line in Fig. 10. At $\lambda = 1.99$, that limiting function is not a continuous smooth curve but has clear abrupt steps. At $\lambda = 2$, the envelope function, represented by a solid line in Fig. 10 is continuous. The lifting of the degeneracy away from integrable points is associated with the opening of discontinuities in the envelope function. An Aubry transition occurs abruptly at $\lambda = 2$ in the commensurate case.

For $\lambda > 2$, the situation is different: the degeneracy is also lifted but the structure selected is different (see Fig. 11, right). A trimer is formed with a sequence of bonds

$$LSSLLSSL \dots, \tag{131}$$

*i.e.* two short ($S$) and two long ($L$) bonds repeating. The points $a_1, a_2$ are also shown in Fig. 8, they are slightly away from the degenerate manifold and at a different place than for $\lambda < 2$. It is also reported in Fig. 9 by the red circles, and corresponds to the $\phi = 0$ structure. The envelope function can be defined as in the previous paragraph and is also discontinuous.

Importantly, in both cases, the continuous degeneracy is lifted and the structure, therefore, cannot be deformed continuously at constant energy. The gap at $k = 0$ (as in Fig. (12), right) is now open as in the 3-gap solution, but, with a difference in the configurations and no continuous degeneracy. Thus, in this generic case away from integrability, we find what one expects from general arguments: a definite set of distortions related by discrete symmetries, while all the gaps are opened and there is no degeneracy: the commensurate CDW is pinned. The integrable case at $\lambda = 2$ thus appears as a special point where not only the two configurations at $\lambda < 2$ and $\lambda > 2$ become degenerate, but, also appears a continuous manifold of degenerate

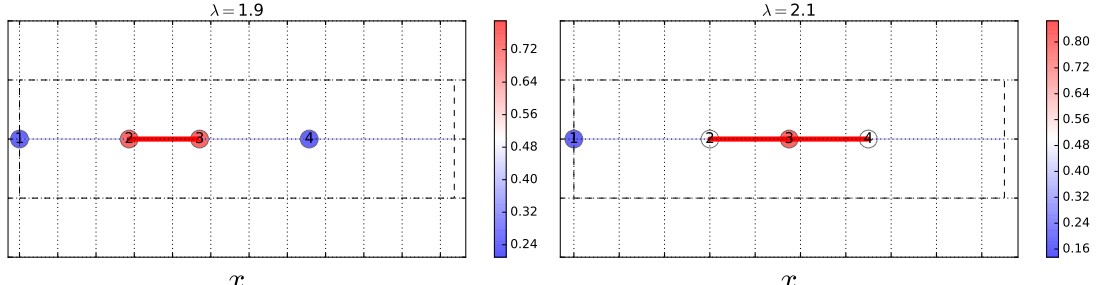

Figure 11: Solution in the nonintegrable case, $\lambda \neq 2$. $\lambda < 2$ (left): the pair of electrons ensures a strong covalent bond between sites 2 and 3 (dimer), other bonds are weaker. $\lambda > 2$ (right): the pair of electron occupies a trimer, other bonds are weaker. The unit-cell (dashed rectangle) with $N = 4$ atoms is periodically repeated. The density of electrons is indicated by the color scale, it is strong in red and weaker in blue. The integrable case appears as a special point where dimer and trimer, as well as continuous configurations between them, are degenerate. Here, $p = 0.02$, $\xi = 2$, $\xi_4 = 0$.

states, thus allowing the structure to be continuously deformed at no energy cost. It has the peculiar property that a *commensurate* charge-density wave achieves Fröhlich conductivity.

To show more clearly the lifting of the continuous degeneracy, we continuously deform the solutions. In this purpose, we fix the length of the strong bond to a given value $\ell$ with $\ell_{min} \leq \ell \leq \ell_{max}$, where $\ell_{min}$ is the length of the strongest bond ($S$) in the ground state and $\ell_{max}$ that of a weaker bond ($L$ or $L_1$), and minimize the energy $W$ numerically. For $\ell = \ell_{min}$ or $\ell = \ell_{max}$, we have $W = W_{min}$. In Fig 13, we plot the difference of energy $W - W_{min}$ as a function of $\ell$. For $\lambda = 2$, we see that changing $\ell$ in the specified range does not change the energy: this is the continuous degeneracy of the integrable case. For $\lambda \neq 2$, the degeneracy is lifted and the energies of the intermediate configurations with $\ell_{min} < \ell < \ell_{max}$ are higher. There is an energy barrier which is necessary to deform the CDW, called the Peierls-Nabarro barrier. Note that it can be weak near $\lambda = 2$, since it continuously vanishes. To interpret its magnitude, it is interesting to compare with the dimer case. Consider three sites: sites 1 and 2 are separated by the vacuum length, $\ell_0$ (defined in Eq. (11)) and site 3 is at a larger distance. What is the energy needed to transfer the first bond onto the second? In the limit when the total length becomes large enough, it is the energy corresponding to the separation of the two bound atoms (using Eq. (10) with $\ell = \ell_0$),

$$W - W_{min} = \frac{1 - \frac{1}{\lambda}}{(\xi\lambda)^{\frac{1}{\lambda-1}}} . \tag{132}$$

For $\lambda = 1.9$ and $\xi = 1$ (as chosen in Fig. 13), one gets $W - W_{min} = 0.2321$, which is considerably larger than the magnitude of the barrier at $p = 0.02$. At larger pressures, when the Peierls gap becomes smaller, the barrier of the three site problem can be considerably smaller. The small amplitude shown in Fig. 13 is thus the result of a combination of large pressures and proximity to the integrable point (where the barrier vanishes).

### 4.3 Example of incommensurate solution: $c = 2 - \varphi$

We now consider an incommensurate filling, which we choose to be $c = \frac{3-\sqrt{5}}{2} = 2 - \varphi$ $\approx 0.38197\ldots$, where $\varphi$ is the golden number. In order to study this case, we approximate $c$ by a sequence of commensurate fractions $r_n/N_n$ obtained from continued fraction:

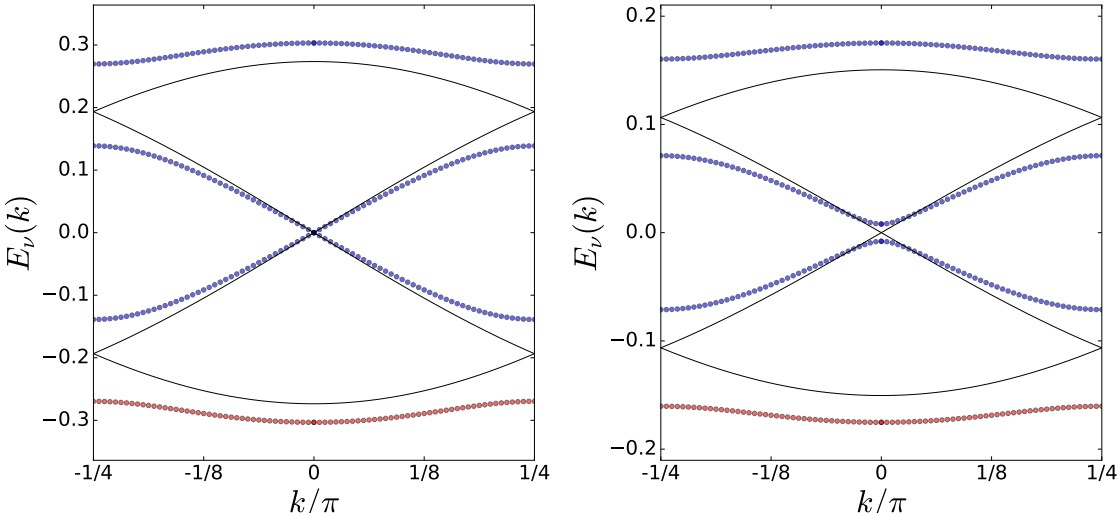

Figure 12: Band structure of the commensurate CDW at quarter filling (filled band showed with red points) for the integrable case $\lambda = 2$ (left) and nonintegrable case $\lambda = 1.6$ (right). The gap at $E = 0$ opens when $\lambda < 2$ and the CDW gets pinned. Here, $p = 0.02$, $\xi = 2$, $\xi_4 = 0$. The black lines are the band structure for the uniform structure with the same averaged length (metallic state), for comparison: not only the Peierls gap opens at the Fermi energy but all energy levels are affected.

$$c \sim \frac{0}{1}, \frac{1}{2}, \frac{1}{3}, \frac{2}{5}, \frac{3}{8}, \frac{5}{13}, \frac{8}{21}, \frac{13}{34}, \frac{21}{55}, \frac{34}{89}, \frac{55}{144}, \cdots \tag{133}$$

This sequence converges to $c$ (Fig. 14), with an error smaller than $1/N_n^2$ [46]. Various chains of period $N_n$ with $r_n$ pairs of electrons will be considered and $N_n \to +\infty$ corresponds to the incommensurate limit. The numerical minimization is done for the BDK Volterra model at $\lambda = 2$ (integrable case) and away from the integrable point at $\lambda \neq 2$.

### 4.3.1 Integrable case

In the integrable case, $\lambda = 2$, we have minimized numerically the energy of various chains with the successive rational approximants of $c$. In Fig. 15 (top), we plot $(\ell_i - \bar{\ell})/\bar{\ell}$, where $\ell_i$ are the optimal distortions obtained and $\bar{\ell}$ the average bond length, versus $ic$ (mod 1) at $p = 0.006$, $\xi = 2$ and for various rational approximants of $c$. When the approximant gets better, the numbers $ir_n/N_n$ (mod 1) fill more and more the $[0, 1]$ segment, allowing to visualize more clearly the envelope function. The envelope function clearly appears continuous. On the other hand, the exact solution for the bond length $\ell_i$ is given by Eq. (99),

$$\ell_i(\phi) = \bar{\ell} + \ln \frac{\vartheta_3((i+1)c + \phi, q)\vartheta_3((i+2)c + \phi, q)}{\vartheta_3(ic + \phi, q)\vartheta_3((i+3)c + \phi, q)}, \tag{134}$$

where $i = 1, \ldots, L$. Here and above, we have chosen a given $\phi$ which reflects the simple possible translations. The averaged bond length $\bar{\ell}$ is known from the numerics. The parameter $q = 0.248$ is deduced from $I_2 = \sum_{i=1}^{N} a_i(\phi)^2$. The envelope function is given by Eq. (134) (shown by the solid line in Fig. 15 (top)). The agreement between numerical results and the analytic result of BDK is excellent (there is no free parameter). It confirms that the 2-gap Ansatz is the ground state in this incommensurate case at integrable points.

The continuity of the envelope function ensures that the energy barriers for the CDW to slide are vanishingly small. Indeed, to transfer the strongest bond (and its electrons) on the

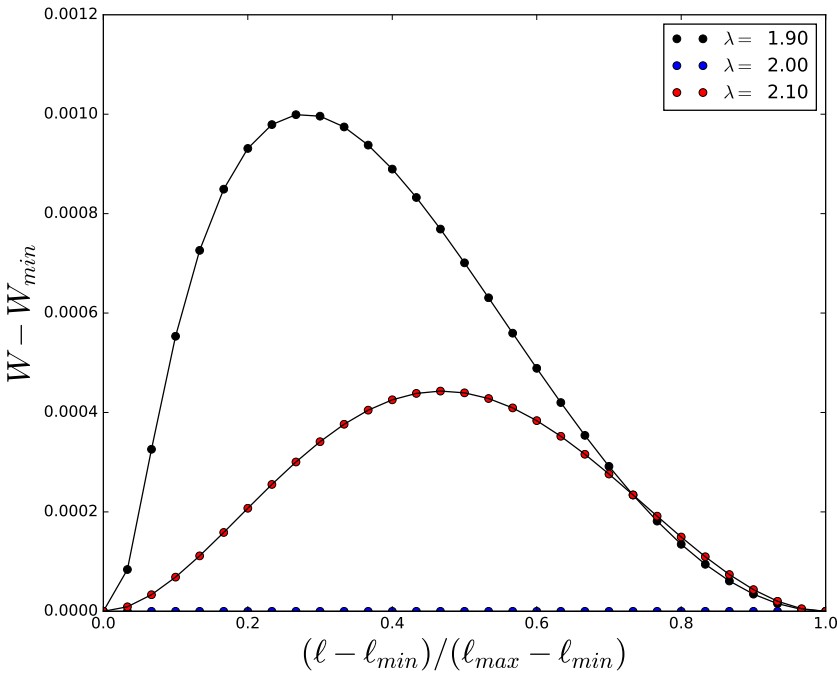

Figure 13: Energy barrier of Peierls-Nabarro to transfer a strong bond (with its electrons) onto the next bond, for $\lambda = 1.9, 2, 2.1$. The barrier vanishes at the integrable point $\lambda = 2$ and the commensurate CDW is unpinned. Here, $p = 0.02$, $\xi = 1$, $\xi_4 = 0$.

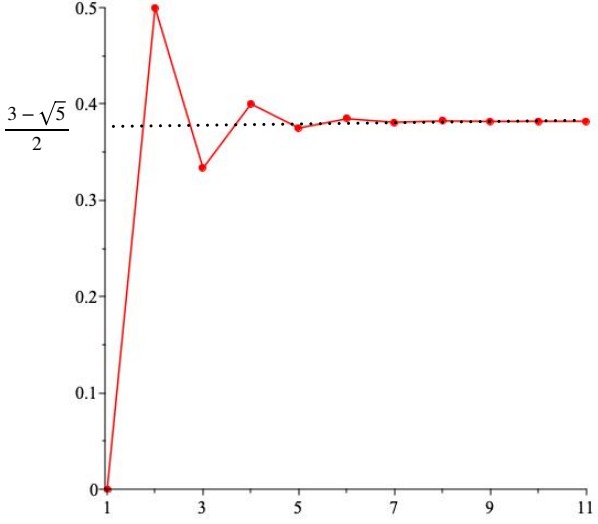

Figure 14: The first rational approximants of $\frac{3-\sqrt{5}}{2}$ given by Eq. (133).

next one, one finds a slightly weaker bond (by a vanishing quantity in the incommensurate limit) somewhere along the chain: a new state, obtained by translating the original state so that the slightly weaker bond is placed on the original strongest bond, has almost the same energy. A very small barrier (vanishingly small in the incommensurate limit) has to be overcome. By iterating this procedure, one understands that there is no energy barrier to elongate the first

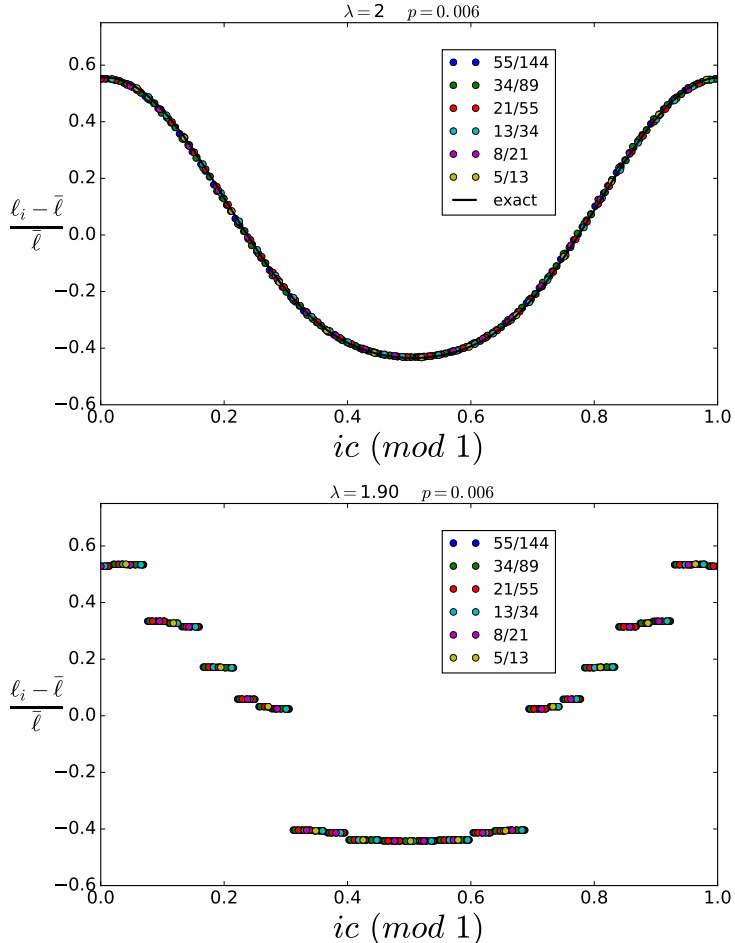

Figure 15: Envelope functions of the optimal bond lengths for the incommensurate CDW with various rational approximants. Top: integrable case ($\lambda = 2$), continuous cnoidal wave. We show the exact result (solid line) from Eq. (134). Bottom: nonintegrable case ($\lambda = 1.9$), discontinuous envelope. Here $p = 0.006$, $\xi = 2$.

bond (and reduce the second bond).

### 4.3.2 Nonintegrable case: Aubry transition

Away from integrability when $\lambda$ is decreased to $\lambda = 1.9$, one sees in Fig. 15 (bottom) that the envelope function is no longer continuous but has gaps at several places. This is the breaking of analyticity, which occurs in other similar models [41, 44, 45]. The chain configuration can be described in the limit of small pressure (*e.g.* $p = 0.006$). For a rational approximant $c = r/N$, there are $r$ short bonds which form distinct local dimers or strong covalent bonds in the unit-cell of $N$ sites (see Fig. (16) when $c = 5/13$). This explains the $r$ lower distinct values of $(\ell_i - \bar{\ell})/\bar{\ell}$ in Fig. 15 (bottom). The dimers occur in a special sequence (see Fig. (16)) which corresponds to a uniform structure (the definition of uniform structures is found in Ref. [59, 60]): this way, the pairs of electrons located on each dimer gain the most energy. This analysis only stands for low enough pressures. When the pressure increases, as shown in Fig. 17, the discontinuities in the envelope function decrease and vanish above some critical pressure. The envelope function becomes continuous in the incommensurate limit and the phase is sliding as in the integrable case. This is an Aubry transition from a high-pressure

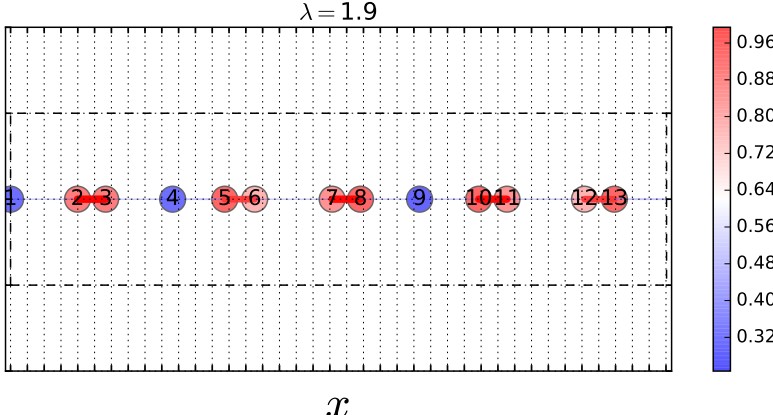

Figure 16: Ground state configuration for a chain of period $N = 13$ filled with 5 pairs of electrons ($c = \frac{5}{13}$), in the nonintegrable case ($\lambda = 1.9$) and low enough pressure (here $p = 0.006$) to be below the Aubry transition: the CDW form local dimers (five strong covalent bonds with the five pairs of electrons). The scale on the right is the electronic density.

unpinned phase to a low-pressure pinned phase. It is similar to that occurring in the SSH model as function of pressure [41, 44] or electric field via an electromechanical effect [45]. The corresponding band structures are shown in Fig. 18: the main Peierls gap does not change much but the secondary gaps are clearly visible at small enough pressures. While the transition is an insulating to metal transition, the electronic gap remains open, but the deformation of the incommensurate CDW leads to Fröhlich conductivity.

The order parameter of the Aubry transition can be taken as the main gap in the envelope function (as in Fig. 17). It is plotted as a function of pressure in Fig. 19. While it shows a smooth crossover for $c = 5/13$, one sees that the crossover is much sharper for $c = 34/89$, which indicates the onset of a transition in the incommensurate limit: the order parameter vanishes linearly with pressure (except for a very slow convergence of the minimization at the transition). The energy is, however, continuous at the transition (see the inset of Fig. 19). The Peierls-Nabarro barrier also vanishes at the transition. The barriers and their evolution as a function of pressure are shown in Fig. 20. As in the commensurate case, the barriers increase slowly in the pinned phase and remains much smaller than the zero-pressure limit.

At strong pressures, the length of the chain must be small and each bond is compressed. It turns out that not only the averaged bond length is small but the distortions around the average $\ell_i - \bar{\ell} = \delta\ell_i$ are small. For example, for $p = 0.1$, the result of the numerical minimization[9] is given in Fig. 21. Both in the integrable and nonintegrable cases, we find a standard cosine CDW. In the integrable case, the exact solution can be expanded when $q \to 0$, by keeping the first term in the Fourier series of $\vartheta_3(z, q)$ [Eq. (85)],

$$\ell_i = \bar{\ell} + u\cos(2\pi i c + \phi), \tag{135}$$

where $u = 8q\sin(\pi c)\sin(2\pi c) \to 0$ and $\phi$ a phase. The Peierls Ansatz is exact in this limit. Since $u \ll \bar{\ell}$, the model can be expanded around a true metallic state and the perturbative approach summarized in section 2.4 is essentially correct.

---

[9]Note that the numerics has to be done carefully, because on small chains, the minimization gives an undistorted metallic state as the ground state. It is necessary to increase the number of unit-cells, *i.e.* the number of $k$ points, for the system to converge to a distorted state, as expected on general grounds [61].

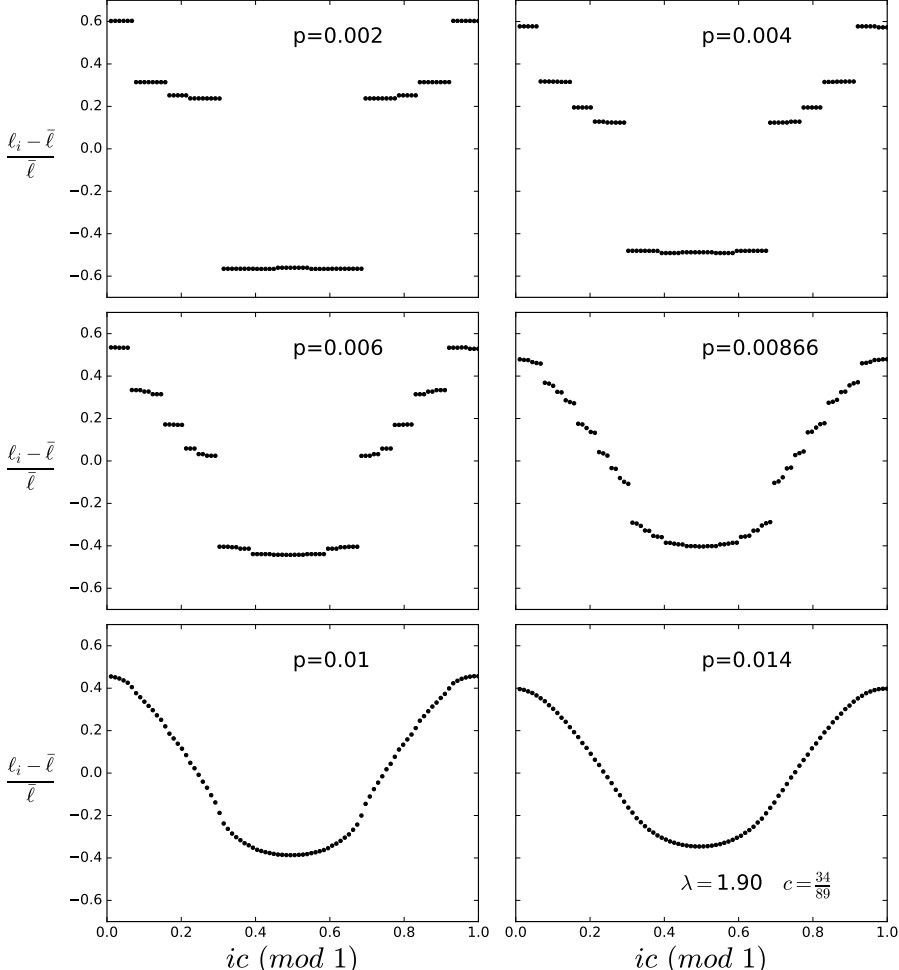

Figure 17: Envelope functions for the $c = \frac{34}{89}$ approximant of the incommensurate case, for different pressures $p$, showing an Aubry transition from a continuous to a discontinuous envelope function. Here $\lambda = 1.90$, $\xi = 2$.

What makes the Peierls approach break down at smaller pressures with the emergence of discontinuities in the envelope function? This is precisely a regime where the nonlinearities in $\delta\ell_i$ become important. When $\delta\ell_i$ get larger, the Hamiltonian acquires a large modulated perturbation. The minimization equations (39) get larger nonlinear terms in $\delta\ell_i$ and define real nonlinear maps. In these maps, the growing nonlinearities are eventually responsible for the destruction of the KAM tori. It is therefore not sufficient to treat the problem in perturbation theory, nor to restrict, as it is done in the standard Peierls approach, the effect of the modulated potential to the energy levels close to the Fermi energy. We have in particular observed that the whole spectrum is modified with respect to that of the metallic state (see *e.g.* Fig. 12). The linear response and the choice of the Peierls Ansatz obviously miss the nonlinear regime.

Although it is difficult to represent the KAM tori in such high-dimensional phase-space, we show some sections of them by plotting $a_{i+1}$ as a function of $a_i$, $i = 1, \dots, N$, in Fig. 22. For $\lambda = 2$ and a given pressure (Fig. 22 (left)), the set of points corresponding to the exact solution forms a regular "trajectory" and corresponds to a section of the Liouville torus on which the solution takes place. When $\lambda = 1.9$ (at the same pressure), one sees that the torus is destroyed. Similarly in Fig. 22, right, one sees the effect of the external pressure at non-integrable points: regular trajectories corresponding to continuous solutions still exist, they

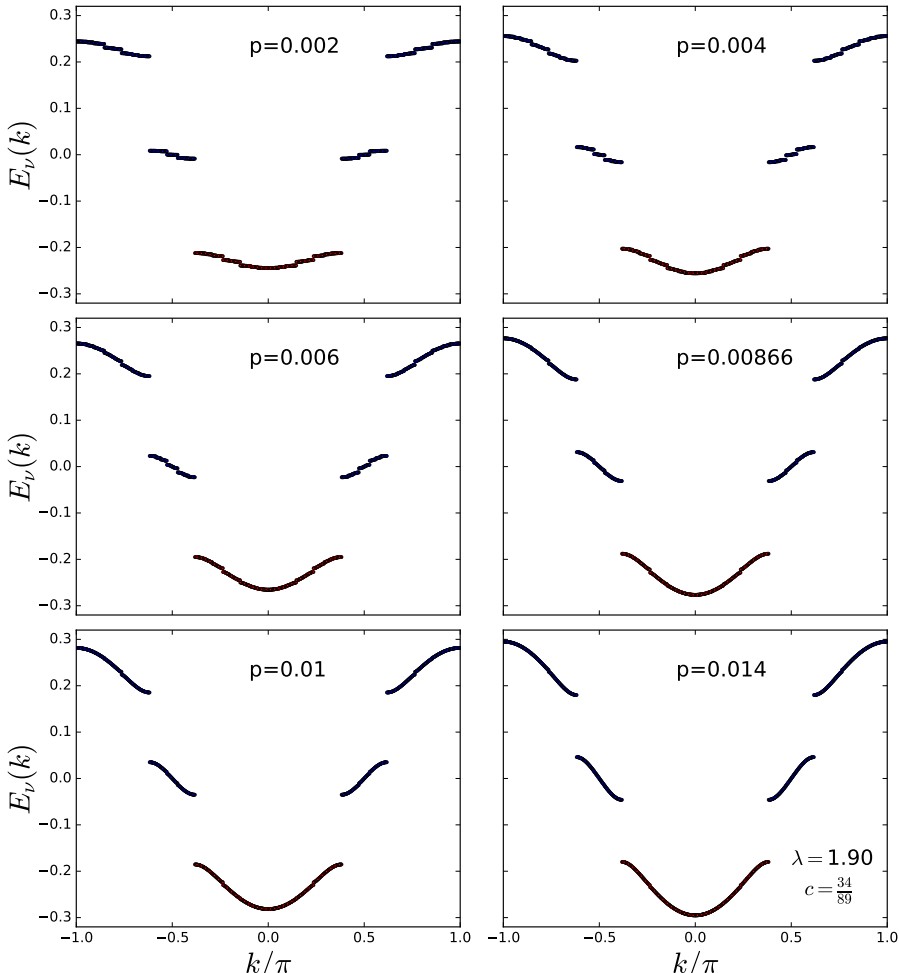

Figure 18: Energy bands in the unfolded Brillouin zone for the $c = \frac{34}{89}$ approximant of the incommensurate case, for different pressures $p$ showing gap openings at low pressures. $\lambda = 1.90$, $\xi = 2$.

are sections of the KAM tori, but eventually at smaller pressures, the KAM tori are destroyed. This is another representation of the Aubry transition, which emphasizes the connection with nonlinear dynamical systems.

It is also important to stress that the SSH Hamiltonian (see Fig. 1) is a linearized version of the BDK model but also exhibits an Aubry transition [41]. The energy and the minimization equations remain nonlinear in the local modulation amplitudes. However, the SSH model breaks down at strong couplings when the distortions become large enough: the hopping amplitude changes sign. The BDK model does not suffer such a setback: the hopping remains well-defined. This allows the chain to remain stable in the weak pressure regime where the averaged bond length and the modulation amplitudes get larger. Thus the chain can be connected to the zero pressure limit, where the atoms form isolated molecules: this is the "anti-integrable" limit (in the language of Aubry and Abramovici [57]) where the physics is local. This limit does not exist in the SSH model.

Note, however, that, when the Aubry transition takes place, the amplitudes of the distortions can be much larger than experimentally observed: this suggests that the intrinsic pinning would occur at too strong energy scales. Pinned CDWs with much smaller amplitudes can be achieved in some other nonlinear models. Indeed, in order to get an Aubry transition, the nonlinearities need to be above a threshold: this does not imply that the distortions should

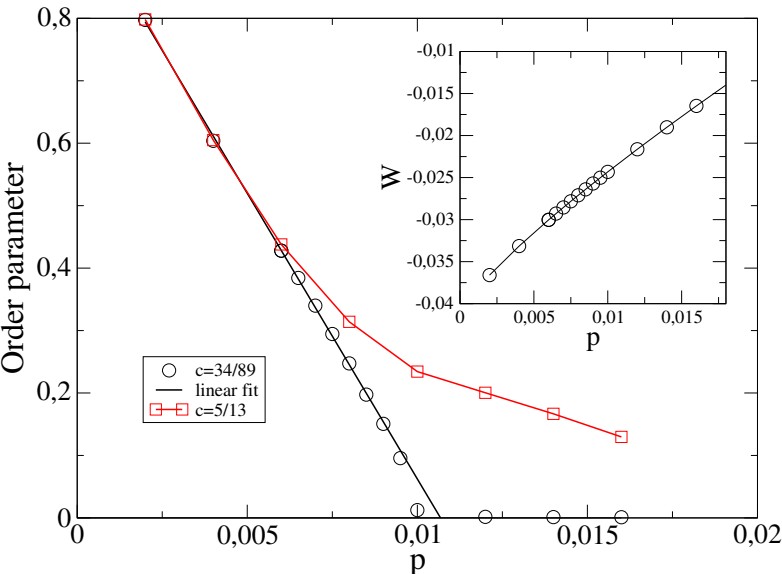

Figure 19: Order parameter of the Aubry transition as a function of pressure defined as the largest gap in the envelope function, see Figs. 17. Inset: no abrupt change in energy nor in its first derivative. Here $\lambda = 1.90$, $\xi = 2$.

be large, only the local gradients. The local gradients can be sufficiently large if the Hamiltonian contains electron-lattice interactions that are strong nonlinear functions of the atomic distance. An example of such a model will be provided in a further publication [62].

# 5  Conclusion

While a 1D Peierls system develops a $2k_F$ charge-density wave at low temperatures, the shape of its envelope function is an interesting issue, both theoretically and experimentally.[10] We have emphasized here that the shape depends on the interaction terms and energy parameters and implies *qualitative* physical consequences.

First, for the BDK model at integrable points, we have confirmed numerically that its ground-state is a cnoidal charge-density wave (1-gap state in the Toda case, 2-gap state in the Volterra case), as originally predicted [31]. The other $g$-gap Ansätze, which develop more zero-energy phason modes, cannot be excluded for a generalized BDK model: we have investigated in details the example of the quarter-filled chain, and found that, although the $g$-gap states could be metastable, the cnoidal wave remains the ground state in that case. The BDK model thus provides an interesting example of a nonlinear model where the shape of the CDW can be explicitly calculated in a nonperturbative regime. It reduces to the Peierls cosine shape in the perturbative regime.

Second, when integrability is broken, we have explained that the commensurate phases are pinned, as expected on general grounds. The absence of pinning at integrable points and for the commensurate phases results from the fine-tuning of the model, which consists of defining a constant energy on a Liouville torus of the underlying classical integrable model.

---

[10]The intensity of higher harmonics are measured in synchrotron X-ray diffractions (see [63]). It is possible to obtain some details about the local atomic structure through the atomic pair distribution function technique [18].

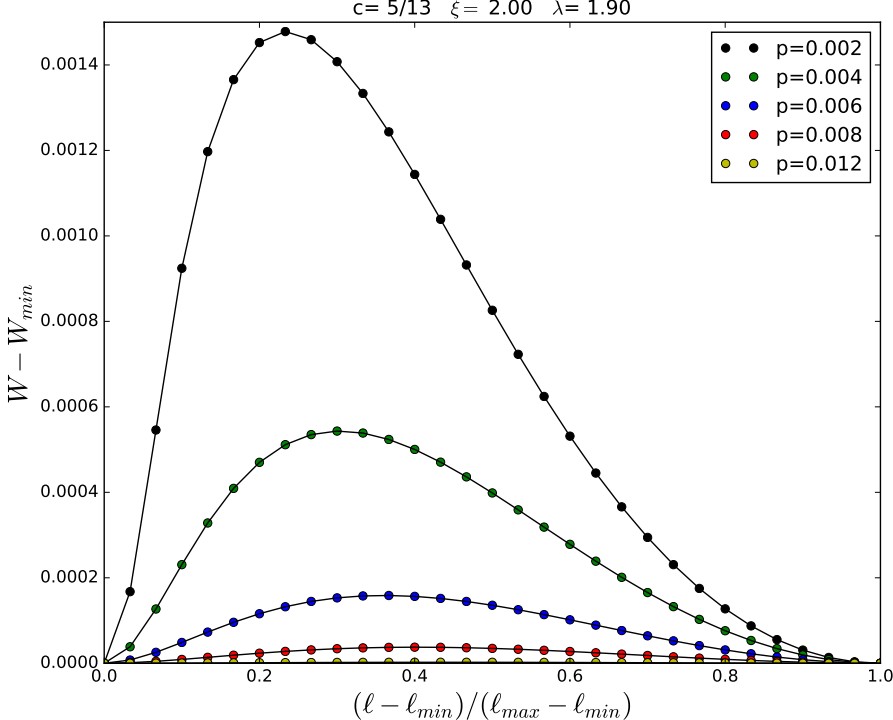

Figure 20: Energy barriers of Peierls-Nabarro for different pressures. There is a pinned-to-unpinned Aubry transition when the barrier vanishes. Here $\lambda = 1.90$, $\xi = 2$.

Now, in the incommensurate case and thanks to KAM theorem, two situations arise:

- At strong pressures, the bonds are contracted and their lengths weakly modulated. The resulting modulated potential implies weak nonlinearities in the ground state equations. There is no fundamental difference with the integrable case and this reflects the KAM theorem in the weak nonlinear regime. The envelope of the wave remains continuous and close to a cnoidal wave. The incommensurate CDW phase is sliding. It corresponds to a dynamical strong-pressure fixed-point and to Peierls-Fröhlich phase.

- At weak pressures, the bonds are elongated and more strongly modulated. The modulated potential being larger, the nonlinearities become important. The system enters the nonlinear stochastic regime in which the KAM tori are broken. The envelope function of the wave is discontinuous. The CDW phase is intrinsically pinned. Atoms tend to form some oligomers (like dimer or trimer local structures): the system has evolved towards an "anti-integrable" limit. It corresponds to a dynamical weak-pressure fixed point. Many metastable states with local defects emerge at higher energies.

In the incommensurate case, an Aubry transition (which is not a crossover) takes place between these two regimes. This is a form of transition to chaos in which Liouville (KAM) tori of the integrable model are destroyed [40]. The resulting pinning of these states appears as an intrinsic nonlinear mechanism. In this case, the Peierls approach breaks down because the modulated potential is strong and must be treated nonperturbatively: the band structure is strongly affected as a whole, even far from the Fermi energy, which is confirmed by some recent experimental results [14]. These states with discontinuous envelope functions could explain the observed charge-density waves and their pinning. Eventually, the exact states with

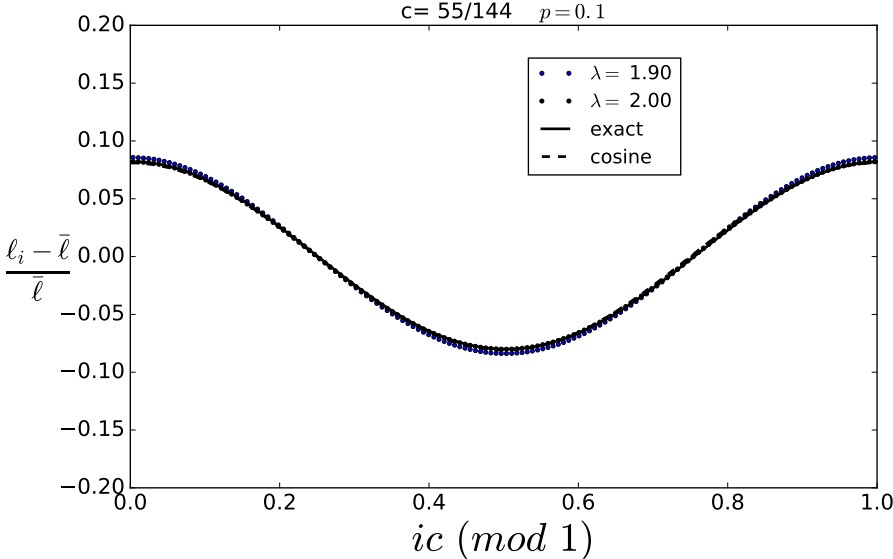

Figure 21: Envelope function of the incommensurate CDW at strong pressure ($p = 0.1$), in the integrable case, $\lambda = 2$ and nonintegrable case $\lambda = 1.9$. Here $\xi = 2$. In full line, exact result from Eq. (134) with $q = 0.0225$ and cosine from Eq. (135).

continuous envelope functions found in the BDK model at integrable points do not exhaust the physical possibilities of Peierls systems.

## Acknowledgments

We would like to thank G. Masbaum from the Institut de Mathématiques de Jussieu (Paris) for stimulating discussions on various mathematical problems related to the present paper.

## A  Equation for the electronic band structure of a 1D problem

How do we compute the energy bands of a 1D periodic tight-binding model with arbitrary nearest neighbor hoppings and potentials? In this appendix, we show that the electronic bands $E_\nu(k)$ satisfy an algebraic equation of the form:

$$Q(E) = \cos kN \,, \tag{A.1}$$

where $N$ is the period and $Q(E)$ a polynomial of degree $N$.

Let us consider a Bloch-wave function with wavevector $k$, amplitude $\psi_i(k)$ at the site $i$ and eigenvalue $E = E_\nu(k) = E_\nu(-k)$. It satisfies the eigenequation for all sites $i$,

$$-a_i\psi_{i+1}(k) - a_{i-1}\psi_{i-1}(k) + b_i\psi_i(k) = E\psi_i(k). \tag{A.2}$$

The successive amplitudes can be obtained by iteration:

$$\begin{pmatrix} \psi_{i+1}(k) \\ \psi_i(k) \end{pmatrix} = \frac{(-1)}{a_i}\begin{pmatrix} E - b_i & a_{i-1} \\ -a_i & 0 \end{pmatrix}\begin{pmatrix} \psi_i(k) \\ \psi_{i-1}(k) \end{pmatrix}. \tag{A.3}$$

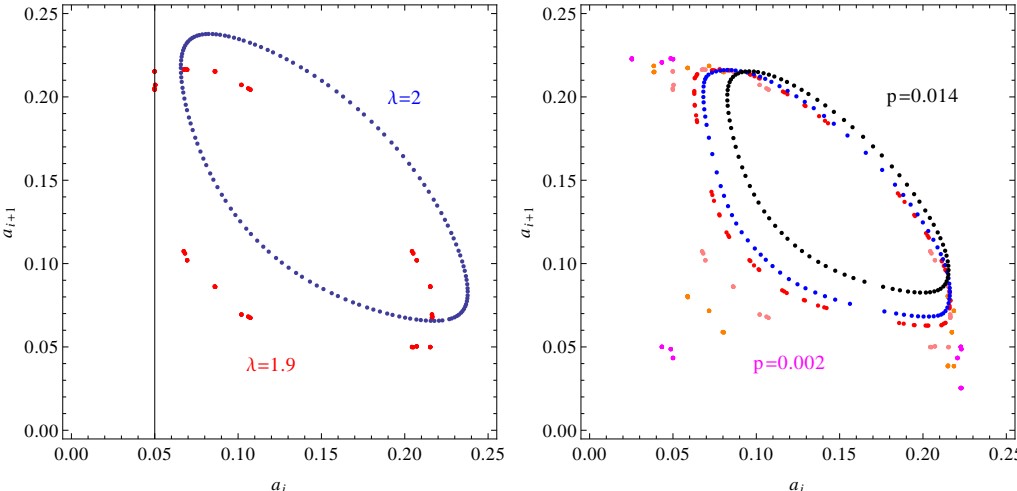

Figure 22: Ground state solutions represented in some sections of the phase-space $(a_{i+1}, a_i)$, $i = 1, \ldots, N$. Left: a section of the Liouville torus ($\lambda = 2$) and its destruction at $\lambda = 1.9$, for $p = 0.006$, $\xi = 2$ and $c = 55/144$. Right: persistence of KAM tori at strong pressures away from integrable points and its eventual destruction at small pressures, with the same data as in Fig. 17.

Starting from $\psi_1(k)$ and $\psi_2(k)$, we iterate the relation over one period $N$, and we get

$$\begin{pmatrix} \psi_{N+2}(k) \\ \psi_{N+1}(k) \end{pmatrix} = \frac{(-1)^N}{C_N} \begin{pmatrix} p_1(E) & p_2(E) \\ p_3(E) & p_4(E) \end{pmatrix} \begin{pmatrix} \psi_2(k) \\ \psi_1(k) \end{pmatrix} \equiv T \begin{pmatrix} \psi_2(k) \\ \psi_1(k) \end{pmatrix}, \qquad \text{(A.4)}$$

where $C_N = \prod_{i=1}^{N} a_i$. $p_i(E)$ are polynomials in $E$ with real coefficients: $p_1(E)$ is of degree $N$, where the coefficient of $E^N$ is 1. $p_4(E)$ is of degree $N-1$. For a Bloch wave function, one has $\psi_{i+N}(k) = e^{ikN} \psi_i(k)$, i.e.

$$\begin{pmatrix} \psi_{N+2}(k) \\ \psi_{N+1}(k) \end{pmatrix} = e^{ikN} \begin{pmatrix} \psi_2(k) \\ \psi_1(k) \end{pmatrix}. \qquad \text{(A.5)}$$

This vector ${}^t(\psi_2(k), \psi_1(k))$ is an eigenvector of the matrix $T$ with eigenvalue $e^{ikN}$.

The same result holds for the wavevector $-k$ and in that case, the eigenvector is ${}^t(\psi_2(-k), \psi_1(-k))$ with the eigenvalue $e^{-ikN}$. Eventually, one gets the two eigenvalues of the $2 \times 2$ matrix $T$, $e^{ikN}$ and $e^{-ikN}$, which gives

$$e^{ikN} + e^{-ikN} = \text{Trace}(T) = 2\cos(kN). \qquad \text{(A.6)}$$

By defining

$$Q(E) = \frac{1}{2}\text{Trace}(T), \qquad \text{(A.7)}$$

one gets Eq. (A.1). On the other hand, half the trace of $T$ is given by

$$Q(E) = \frac{(-1)^N}{2C_N}(p_1(E) + p_4(E)) = A_0(E^N - I_1 E^{N-1} + \cdots - I_N), \qquad \text{(A.8)}$$

which is Eq. (21).

# B  Energy gradients and nonlinear minimization equations

We calculate the gradient of the energy with respect to the classical variables $\{a_i\}$ (local hoppings) and $\{b_i\}$ (local potentials) and obtain the minimization equations.

In the Hamiltonian (9), let $i$ be an arbitrary site, where $b_i$ is changed into $b_i + \delta b_i$, let other $b_j$, $j \neq i$, unchanged. At first order, one gets the following perturbation:

$$\delta H/t = \delta b_i \sum_\sigma c_{i\sigma}^\dagger c_{i\sigma} + 2\xi b_i \delta b_i \,. \tag{B.1}$$

At first order in perturbation theory, it gives an energy correction $\delta W = \langle \delta H \rangle/(2St)$,

$$\delta W = \frac{1}{2S} \delta b_i \sum_\sigma \langle c_{i\sigma}^\dagger c_{i\sigma} \rangle + \frac{\xi}{S} b_i \delta b_i \,. \tag{B.2}$$

The gradient is

$$\frac{\delta W}{\delta b_i} = \frac{1}{2S} \sum_\sigma \langle c_{i\sigma}^\dagger c_{i\sigma} \rangle + \frac{\xi}{S} b_i \,, \tag{B.3}$$

and the minimization equations $\delta W/\delta b_i = 0$ write

$$2\xi b_i = -\sum_\sigma \langle c_{i\sigma}^\dagger c_{i\sigma} \rangle \,, \tag{B.4}$$

where $\sum_\sigma \langle c_{i\sigma}^\dagger c_{i\sigma} \rangle \equiv n_i$ is the electronic density at site $i$. The linear response consists of linearizing these equations, for example, when the instability of a metal is studied. In this case, using $b_j = \bar{b} + \delta b_j$, one has $n_i = n_i^{(0)} + \sum_j \delta b_j \frac{\partial n_i}{\partial b_j} \equiv n_i^{(0)} + \sum_j \chi_{ij} \delta b_j$ where $\chi_{ij}$ is the electronic susceptibility of the metal. In general, these equations are coupled nonlinear equations, the right-hand-side being a nonlinear function of $\{a_j\}$, $\{b_j\}$. The aim of the present paper is to point out the effects of the nonlinearities and solve the nonlinear equations.

Similarly for the hopping variables $a_i = \frac{1}{2} e^{-\frac{\ell_i}{2}}$, or the bond length $\ell_i$, consider a change $\ell_i \to \ell_i + \delta \ell_i$ for a site $i$, leaving $\ell_j$, $j \neq i$ unchanged. The corresponding energy change at first-order is

$$\delta W = \frac{1}{8S} e^{-\frac{\ell_i}{2}} \delta \ell_i \sum_\sigma \langle c_{i+1\sigma}^\dagger c_{i\sigma} + \text{h.c} \rangle - \frac{\lambda}{2S} \xi (\frac{1}{2} e^{-\frac{\ell_i}{2}})^\lambda \delta \ell_i + \frac{p}{2S} \delta \ell_i \,, \tag{B.5}$$

and the gradient writes

$$\frac{\delta W}{\delta \ell_i} = \frac{1}{4S} a_i \sum_\sigma \langle c_{i+1\sigma}^\dagger c_{i\sigma} + \text{h.c} \rangle - \frac{\lambda}{2S} \xi a_i^{\lambda} + \frac{p}{2S} \,. \tag{B.6}$$

The extrema should satisfy $\delta W/\delta \ell_i = 0$, i.e.

$$\lambda \xi a_i^{\lambda} = \frac{1}{2} a_i \sum_\sigma \langle c_{i+1\sigma}^\dagger c_{i\sigma} + \text{h.c} \rangle + p \,, \tag{B.7}$$

which involve the bond charge $\sum_\sigma \langle c_{i+1\sigma}^\dagger c_{i\sigma} + \text{h.c} \rangle$, which is also a nonlinear function of the classical variables. Eqs. (B.7) and (B.4) form coupled nonlinear equations for $\{a_i\}$ (or $\{\ell_i\}$) and $\{b_i\}$.

# C  Energy gradients, explicit example of $c = 1/4$

We illustrate the derivation of the energy gradients $\delta W/\delta I_m$ in the Volterra case at quarter-filling, $c = 1/4$, for which the periodicity of the chain is $N = 4$.

We have seen that to obtain the energy bands and the characteristic polynomial of $H(k)$, it is convenient to define the polynomial $Q(E)$, Eq. (21), which writes, for $N = 4$,

$$Q(E) = A_0(E^4 - I_2 E^2 - I_4). \tag{C.1}$$

It only involves even powers of $E$ (giving a $E \to -E$ symmetry), so the only nonzero $I_m$ terms are

$$
\begin{aligned}
I_2 &= a_1{}^2 + a_2{}^2 + a_3{}^2 + a_4{}^2, &\tag{C.2}\\
I_4 &= -a_2{}^2 a_4{}^2 - a_1{}^2 a_3{}^2, &\tag{C.3}\\
C_4 &= a_1 a_2 a_3 a_4. &\tag{C.4}
\end{aligned}
$$

We use also $A_0 = \frac{1}{2C_4}$ and $C_4 = \frac{1}{16}e^{-\frac{l_0}{2}}$. A generic example of plot of $Q(E)$ is given in Fig. (3) (a). Because the model makes sense only when $\ell_i > 0$, i.e. $a_i = \frac{1}{2}e^{-\frac{\ell_i}{2}} < \frac{1}{2}$, the coefficients follow

$$0 < I_2 < 1, \qquad -\frac{1}{8} < I_4 < 0, \qquad 0 < C_4 < \frac{1}{16}. \tag{C.5}$$

For $N = 4$, we can explicitly solve the algebraic equation,

$$Q(E) = \cos kN. \tag{C.6}$$

The solutions are the four energy bands (supposing $I_2^2 \geq 4(-I_4 + 2C_4)$):

$$E_\nu(k) = \pm\sqrt{\frac{1}{2}\left(I_2 \pm \sqrt{I_2^2 + 4(I_4 + 2C_4 \cos 4k)}\right)}. \tag{C.7}$$

A plot of the energy bands is given in Fig. 3 (b): the four bands are separated by three gaps. They are symmetric with respect to $E = 0$, thanks to the special symmetry mentioned above. Note that the energy bands are expressed in terms of $I_2, I_4, C_4$.

At $c = 1/4$, the lowest band ($\nu = 1$) is filled and the others empty. One wants to minimize the generalized BDK energy,

$$W = \frac{1}{S}\sum_k E_1(k) + \sum_{m=0}^{2} \xi_{2m} I_{2m}, \tag{C.8}$$

The model parameters are $\xi_0 = \frac{1}{2}p$, $\xi_2 = \xi$ and $\xi_4$. The gradient of the energy writes

$$\delta W = \frac{1}{S}\sum_k \delta E_1(k) + \sum_{m=0}^{2} \xi_{2m}\delta I_{2m}. \tag{C.9}$$

We follow the same procedure than that leading to Eq. (50) and get

$$\frac{1}{S}\sum_k \delta E_1(k) = \sum_{m=0}^{2} J_{2m}\delta I_{2m}, \tag{C.10}$$

where

$$J_0 = -\frac{1}{2S}\sum_k \frac{Q(E_1(k))}{Q'(E_1(k))}, \tag{C.11}$$

$$J_2 = \frac{1}{S}\sum_k \frac{A_0 E_1(k)^2}{Q'(E_1(k))}, \tag{C.12}$$

$$J_4 = -\frac{1}{S}\sum_k \frac{A_0}{Q'(E_1(k))}, \tag{C.13}$$

which are sums over $k$ in the first Brillouin zone. $J_0, J_2$ and $J_4$ are functions of $I_0, I_2$ and $I_4$. With these definitions, Eq. (C.9) writes

$$\delta W = \sum_{m=0}^{2} (J_{2m} + \xi_{2m})\delta I_{2m}, \tag{C.14}$$

Eventually, the energy gradients are given by

$$\frac{\delta W}{\delta I_{2m}} = J_{2m} + \xi_{2m}, \qquad \forall m = 0, 1, 2. \tag{C.15}$$

## D  Energy gradients for $g$-gap Ansätze

In this appendix, we find the expression of the gradient of the energy of the integrable model for the special $g$-gap Ansätze. The general expression Eq. (50) can be expressed in terms of the remaining independent $\delta I_m$, leading to Eq. (64).

Suppose that the spectrum has $g$ gaps (instead of $N-1$), which means that $N-1-g$ gaps are accidentally closed. In other words, the polynomial of degree $2N$,

$$R_{2N}(E) = 1 - Q(E)^2, \tag{D.1}$$

the zeros of which gives the energy of the band edges, has $N-1-g$ double roots, $e_i$ (which correspond to degenerate band edges). It can be written,

$$R_{2N}(E) = P_{2g+2}(E) \prod_{i=1}^{N-1-g} (E - e_i)^2, \tag{D.2}$$

where $P_{2g+2}(E)$ is a polynomial of degree $2g + 2$.

On the other hand, each closed gap (corresponding to a double root $e_i$) gives a relation of the form (Eq. 61),

$$\sum_{m=0}^{N} l_m(e_i)\delta I_m = 0, \tag{D.3}$$

where

$$l_0(E) \equiv -\frac{1}{2}Q(E), \tag{D.4}$$

$$l_m(E) \equiv A_0 E^{N-m}, \qquad \forall m \geq 1. \tag{D.5}$$

The double roots can then be factorized out of the polynomial,

$$\sum_{m=0}^{N} l_m(E)\delta I_m = \sum_{m=0}^{g+1} \delta A_m E^m \prod_{i=1}^{N-1-g} (E - e_i), \tag{D.6}$$

where the $\delta A_m$ can be explicitly obtained by expanding and identifying the coefficients of the successive powers of $E$, from $E^N$, $E^{N-1}$ down to $E^{N-g-1}$,

$$-\frac{1}{2}A_0\delta I_0 \;=\; \delta A_{g+1}\,, \tag{D.7}$$

$$A_0\delta I_1 - \frac{1}{2}A_0 I_1 \delta I_0 \;=\; \delta A_g - \delta A_{g+1}\sum_{j=1}^{N-g-1}e_j\,, \tag{D.8}$$

$$\cdots$$

$$A_0\delta I_{g+1} - \frac{1}{2}A_0 I_{g+1}\delta I_0 \;=\; \sum_{m=0}^{g+1}\alpha_m\delta A_m\,. \tag{D.9}$$

This system of $g+2$ equations is directly invertible thanks to its triangular form. For example, one finds that $\delta A_{g+1}$ is simply proportional to $\delta I_0$ from the upper equation. Then, from the second equation, $\delta A_g$ is a linear combination of $\delta I_0$ and $\delta I_1$, etc. down to $\delta A_0$, which is a linear combination of all $\delta I_m$ from $m=0$ to $g+1$. One finds that

$$\delta A_m = \sum_{j=0}^{g+1}\beta_{m,j}\delta I_j\,, \tag{D.10}$$

for $m = 0,\ldots,g+1$. $\delta A_m$ are thus linear combinations of, at most, $g+2$ independent $\delta I_m$. We have $\beta_{m,j} = 0$ for $j+m > g+1$ and, using $s_1 = \sum_j e_j$ and $s_2 = \sum_{i\neq j}e_i e_j$,

$$\beta_{g+1,0} \;=\; -\frac{1}{2}A_0\,, \tag{D.11}$$

$$\beta_{g,0} \;=\; -\frac{1}{2}A_0(I_1 + s_1)\,, \quad \beta_{g,1} = A_0\,, \tag{D.12}$$

$$\beta_{g-1,0} \;=\; -\frac{1}{2}A_0[(I_1 + s_1)s_1 + s_2 + I_2]\,, \quad \beta_{g-1,1} = A_0 s_1\,, \quad \beta_{g-1,2} = A_0\,, \text{ etc. (D.13)}$$

When the system has only $g$ gaps, the gradient of the electronic energy, given by Eq. (50), simplifies, thanks to Eqs. (D.2) and (D.6), into

$$\delta E_{elec} = \int dE\,\frac{\sum_{m=0}^{g+1}\delta A_m E^m}{\sqrt{P_{2g+2}(E)}}\,. \tag{D.14}$$

By using Eq. (D.10), we get (renote the indices)

$$\delta E_{elec} = \sum_{m=0}^{g+1}G_m\delta I_m\,, \tag{D.15}$$

where

$$G_m \;=\; \int dE\,\frac{g_m(E)}{\sqrt{P_{2g+2}(E)}}\,, \tag{D.16}$$

$$g_m(E) \;=\; \sum_{j=0}^{g+1}\beta_{j,m}E^j\,, \tag{D.17}$$

which is the expression given in Eq. (64). If the number of gaps corresponds to the generic case, $g = N-1$, the expression reduces to (50). When $g < N-1$, thanks to the dependency relation, the gradient (D.15) can be expressed as a sum over the $g+2$ independent variables $\delta I_m$ only.

# E Numerical gradient descent algorithm used for the minimization

The numerical minimization is a standard gradient descent algorithm. Suppose that you want to minimize a function $W$ of some variables $\{x_i\}$. The steepest descent consists in updating the variables *downhill*, by a finite amount in the negative direction of the local gradients. For this, one needs to compute the gradients $\delta W/\delta x_i$ at each step and iterate the equations,

$$x_i' = x_i - \frac{\delta W}{\delta x_i} \delta t \,, \tag{E.1}$$

where $\delta t$ is a small numerical increment, until the gradient is smaller than a given threshold. It is equivalent to solving a first-order (overdamped) dynamical equation of motion

$$\frac{\partial x_i}{\partial t} = -\frac{\delta W}{\delta x_i} \,. \tag{E.2}$$

While this method allows one to find a local minimum, it is possible to find distinct minima starting from random configurations, *i.e.* exploring the phase space.

In the present problem, one needs to minimize the energy $W$ which is a function of $\{a_i\}$ (or $\{\ell_i\}$) and possibly $\{b_i\}$. The respective gradients $\delta W/\delta b_i$ and $\delta W/\delta \ell_i$ are calculated in Appendix B. Numerically, they are computed as follows. Let us introduce the Bloch wave function $|\Psi_\nu(k)\rangle$ with mode $\nu$ having energy $E_\nu(k)$ and complex amplitude $\psi_{i,\nu}(k)$, one gets

$$\sum_\sigma \langle c_{i\sigma}^\dagger c_{i\sigma} \rangle = 2 \sum_{k,\nu_{occ.}} |\psi_{i,\nu}(k)|^2 \,, \tag{E.3}$$

where the factor 2 comes from the spin degeneracy. The sum is up to the Fermi energy (the sums over $k$ are performed numerically and using the $k \to -k$ symmetry). The gradient (B.3) writes

$$\frac{\delta W}{\delta b_i} = \frac{1}{S} \sum_{k,\nu_{occ.}} |\psi_{i,\nu}(k)|^2 + \frac{\xi}{S} b_i \,. \tag{E.4}$$

Similarly for the second gradient (B.6), one finds

$$\frac{\delta W}{\delta \ell_i} = \frac{1}{S} a_i \sum_{k,\nu_{occ.}} \text{Re}[\psi_{i+1,\nu}^*(k)\psi_{i,\nu}(k)] - \frac{\lambda}{2S} \xi a_i{}^\lambda + \frac{p}{2S} \,. \tag{E.5}$$

These quantities are computed by diagonalizing the matrix. To get the local extremum, the variables are updated according to

$$b_i' = b_i - \frac{\delta W}{\delta b_i} \delta t \,, \tag{E.6}$$

$$\ell_i' = \ell_i - \frac{\delta W}{\delta \ell_i} \delta t \,, \tag{E.7}$$

until the gradients are smaller than a given threshold, typically $10^{-7}$. The number of iterations needed to reach this precision depends on the parameters, and varies from $10^4$ to $10^6$ near the Aubry transition where the convergence is slow.

# F    Definitions of Jacobi $\vartheta_n(z,q)$-functions

Jacobi $\vartheta_n(z,q)$-functions are important analytic functions which play a special role in the theory of elliptic functions (see, for instance, Ref. [64]). In particular, Jacobi $\vartheta_3(z,q)$ function is defined by its Fourier series,

$$\vartheta_3(z,q) = \sum_{n=-\infty}^{+\infty} e^{2i\pi nz} q^{n^2} = 1 + 2\sum_{n=1}^{+\infty} q^{n^2} \cos(2\pi nz), \tag{F.1}$$

with $z \in \mathbb{C}$. $q$ is a complex parameter, called the nome, and such that $|q| < 1$ to ensure that the series converges. The parameter $q$ is sometimes noted $q = e^{i\pi\tau}$, where $\tau$ is a complex number in the upper half-plane, $\mathfrak{Im}(\tau) > 0$, so that $|q| < 1$. Higher harmonics become rapidly small when $|q| \to 0$. By a direct calculation, one finds that, for $n$ and $m$ integers,

$$\vartheta_3(z+n,q) = \vartheta_3(z,q), \tag{F.2}$$

$$\vartheta_3(z+m\tau,q) = e^{-i\pi m^2\tau - 2i\pi mz}\vartheta_3(z,q). \tag{F.3}$$

The function $\vartheta_3$ is periodic with period 1 (by definition) but not elliptic. One finds that $\vartheta_3(\frac{1+\tau}{2},q) = 0$ and that the only zeros of the function are off the real axis (because the imaginary part of $\tau$ is strictly nonzero).

In this paper, we are only interested in $z \in \mathbb{R}$ and $q \in \mathbb{R}$. Some examples of plots of $\vartheta_3(z,q)$ are given in Fig. 23 for $q = 0, 0.1, 0.5, 0.9$.

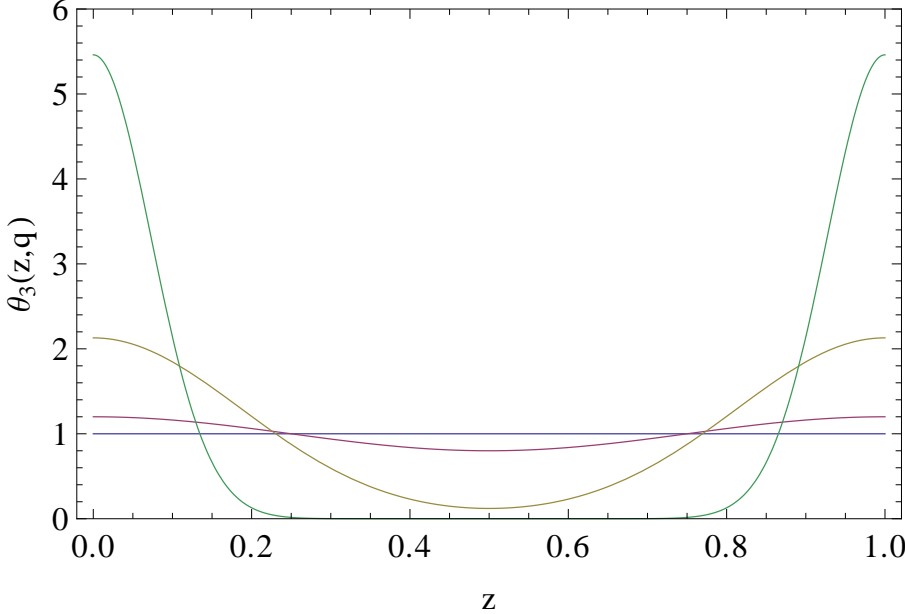

Figure 23: $\vartheta_3(z,q)$ periodic function (with period 1) for $z \in \mathbb{R}$ and $q = 0, 0.1, 0.5, 0.9$.

When $q$ is small, one finds

$$\vartheta_3(z,q) = 1 + 2q\cos(2\pi nz) + \mathcal{O}(q^4), \tag{F.4}$$

and so $\vartheta_3$ is very close to a cosine modulation with amplitude $2q$. When $q$ increases, the modulation amplitude around 1 increases, and the shape is deformed, becoming more and more localized around zero (and, by periodicity, around any integer). $\vartheta_3(z,q)$ never strictly vanishes for $z \in \mathbb{R}$, since its zeros are in the complex plane.

Other Jacobi $\vartheta_n(z,q)$-functions are defined by

$$
\begin{aligned}
\vartheta_1(z,q) &= e^{i\pi\tau/4+i\pi(z+1/2)}\vartheta_3(z+\frac{1}{2}+\frac{\tau}{2},q) \\
&= q^{\frac{1}{4}}e^{i\pi(z+1/2)}[1+2\sum_{n=1}^{+\infty}(-1)^n q^{n^2+n}\cos(2\pi nz)], \quad\quad (\text{F.5})\\
\vartheta_2(z,q) &= e^{i\pi\tau/4+i\pi z}\vartheta_3(z+\frac{\tau}{2},q), \quad\quad (\text{F.6})\\
\vartheta_4(z,q) &= \vartheta_3(z+\frac{1}{2},q)=1+2\sum_{n=1}^{+\infty}(-1)^n q^{n^2}\cos(2\pi nz). \quad\quad (\text{F.7})
\end{aligned}
$$

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
