# Peer review of "On the intrinsic pinning and shape of charge-density waves in 1D Peierls systems"

_SciPost Physics, doi:SciPost Phys. 14, 051 (2023)_

## Round 1 · Referee Report · Anonymous (Referee 1) · 2022-6-27

Strengths

Pedagogical presentation of the state of the art about charge density wave pinning in various models for one-dimension lattice

Weaknesses

Difficult to identify novelties among the quantity of results that are listed from the long history of charge density wave.

Report

I propose to accept the publication on the condition presented below.

Requested changes

The authors must clarify their breakthrough in the field of charge density waves in their abstract and introduction.

Attachment

  • validity: high
  • significance: high
  • originality: ok
  • clarity: top
  • formatting: reasonable
  • grammar: perfect

Author:  Olivier Cépas  on 2022-10-12  [id 2916]

(in reply to Report 1 on 2022-06-27)

We would like to thank the referee for his very positive report and careful reading.

Regarding the weakness pointed out, we have tried to improve the
manuscript by rewritting part of the abstract and introduction. We
emphasize that the paper is not a review in that it includes new
arguments about the old solution of Brazovskii et al. and a thorough study
of a new model which is the perturbed BDK model. This leads
to an Aubry transition, not discussed earlier in this context.

We are grateful to the referee for recognizing the pedagogical
character of the paper, which took us some time to develop and which we hope to be useful.

---

## Round 1 · Referee Report · Anonymous (Referee 2) · 2022-8-31

Strengths

1- Pedagogical course on commensurate/incommensurate one-dimensional systems and on the corresponding work of Brazovskii et al. 2- Scientific global reliance. 3- New numerical results allowing the determination of a phase diagram, which is the real interest of the article. 4- Clarity of the text (except for some details to be modified). 5- The subject is acknowledging a revival so this article may have some visibility if it is published.

Weaknesses

1- Unusual balance between recalled results already well-established and new ones. 2- Irregularity of style, mixing colloquial expressions and pompous ones, telegraphic style and long sentences. 3- Too many repetitions at all orders (inside paragraphs, inside sections and in the whole text). 4- A few points to be clarified, on the mathematical (or physical) plan. Some details must simply be discarded. 5- The length of the article may discourage some readers.

Report

See file rap-CepasQuemerais.pdf

Requested changes

See corrCepasQuemerais.pdf, which link is at the end of rap-CepasQuemerais.pdf. SciPost recommends not to use annotations but, seeing the amount of problems I have found no other method (in particular, I have estimated the time needed to type these annotations to 40 hours). A contradictory injonction for referees is to be constructive, which I have been as much as possible.

Attachment

  • validity: good
  • significance: low
  • originality: low
  • clarity: good
  • formatting: mediocre
  • grammar: mediocre

Author:  Olivier Cépas  on 2022-10-12  [id 2917]

(in reply to Report 2 on 2022-08-31)

See attached file.

Attachment:

reply.pdf

---

## Round 2 · Referee Report · Anonymous (Referee 2) · 2022-10-19

Strengths

1- Pedagogical explanations, 2- new phase diagram for an important family of Hamiltonians 3- innovant numerical investigations

Weaknesses

1-Long article that can discourage reader 2- Ratio of new results which can be disappointing

Report

The authors have addressed most of my concerns and taken into account most of my advices. I had misunderstood some explanations, which have become clear now. About the language, I only tried, in my first report, to help improving the manuscript; it is now satisfactory.
About the balance between afresh presentation of already established results and new numerical ones, I do not fully agree with the authors’ response. They claim that they “reformulate” previous results in a way that is both “original” and “new”. I agree with the originality of the presentation but “new” has a specific meaning, in article reports, which cannot apply here. I have actually not contested the pedagogical rewriting of previous works but only claimed that it is not obvious, reading the introduction, to discriminate the results which are new from others. The introduction has been improved and, if the editor agrees, I think this revised manuscript is worth being published.

Requested changes

I still claim that “whatever”, which is found several times in the manuscript as pronouns or adverb, must be used with a verb, as in “whatever this is”, when one writes in formal (elevated) English, even though the verb is often familiarly omitted.

---

## Round 2 · Author Response

Dear Editor,
We would like to resubmit our paper to SciPost Physics, after having made some major revisions of the text, following referees' advices. In particular, we have clarified ``what is new''. The paper considerably simplifies the solution of the integrable model and emphasizes its limitations. This is only one part of the paper and we are happy that both referees have acknowledged its pedagogical clarity. We stress that the whole study concerning the Aubry transition when the model is made nonintegrable is completely new in this context and is the main topic of the present paper. We thus hope that you will be in a position to accept the present paper without further review, given that the two referee reports are positive and we have included their suggestions.

The authors.

---

## Round 2 · List of Changes

Changes have been made, following the second referee's detailed corrections. They concern essentially the text, some formulations and the english: almost all the suggested changes have been kept (except for a few details such as ``the SSH model'' where ``the'' is kept, as explained in our reply to the referee). We have changed the second part of the abstract and part of the introduction (especially the last paragraph) to emphasize what is new and answer the referees' concerns. Appendix D has been clarified to answer the second referee's mathematical question.

---

## Editorial Decision

published